# Contextualized Messages Boost Graph Representations

## Abstract

Graph neural networks (GNNs) have gained significant attention in recent years for their ability to process data that may be represented as graphs. This has prompted several studies to explore their representational capability based on the graph isomorphism task. These works inherently assume a countable node feature representation, potentially limiting their applicability. Interestingly, only a few study GNNs with uncountable node feature representation. In the paper, a novel perspective on the representational capability of GNNs is investigated across all levels—node-level, neighborhood-level, and graph-level—when the space of node feature representation is uncountable. More specifically, the strict injective and metric requirements are *softly* relaxed by employing a *pseudometric* distance on the space of input to create a *soft-injective* function such that distinct inputs may produce *similar* outputs if and only if the *pseudometric* deems the inputs to be sufficiently *similar* on some representation. As a consequence, a simple and computationally efficient *soft-isomorphic* relational graph convolution network (SIR-GCN) that emphasizes the contextualized transformation of neighborhood feature representations via *anisotropic* and *dynamic* message functions is proposed. A mathematical discussion on the relationship between SIR-GCN and widely used GNNs is then laid out to put the contribution into context, establishing SIR-GCN as a generalization of classical GNN methodologies. Experiments on synthetic and benchmark datasets then demonstrate the relative superiority of SIR-GCN, outperforming comparable models in node and graph property prediction tasks.

## 1 Introduction

Graph neural networks (GNNs) constitute a class of deep learning models designed to process data that may be represented as graphs. These models are well-suited for node, edge, and graph property prediction tasks across various domains including social networks, molecular graphs, and biological networks, among others (Dwivedi et al., 2023; Hu et al., 2020). GNNs predominantly follow the message-passing scheme wherein each node aggregates the feature representation of its neighbors and combines them to create an updated node feature representation (Gilmer et al., 2017; Xu et al., 2018a;b). This allows the model to encapsulate both the network structure and the broader node contexts. Moreover, a graph readout function is employed to pool the individual node feature representation and create a representation for the entire graph (Li et al., 2015; Murphy et al., 2019; Xu et al., 2018a; Ying et al., 2018).

Among the most widely used GNNs in literature include the graph convolution network (GCN) (Kipf & Welling, 2016), graph sample and aggregate (GraphSAGE) (Hamilton et al., 2017), graph attention network (GAT) (Brody et al., 2021; Veličković et al., 2017), and graph isomorphism network (GIN) (Xu et al., 2018a) which largely fall under the message-passing neural network (MPNN) (Gilmer et al., 2017) framework. These models have gained popularity due to their simplicity and remarkable performance across various applications (Dwivedi et al., 2023; Hsu et al., 2021; Hu et al., 2020; Jiang et al., 2022; Kim & Ye, 2020; Liu et al., 2020). Improvements are also constantly being proposed to achieve state-of-the-art performance (Bodnar et al., 2021; Bouritsas et al., 2022; Ishiguro et al., 2019; Miao et al., 2022; Sun et al., 2020; Wang et al., 2019b; Ying et al., 2021).

Notably, these advances are mainly driven by heuristics and empirical results. Nonetheless, several studies have also begun exploring the representational capability of GNNs (Azizian & Lelarge, 2020;

Bodnar et al., 2021; Böker et al., 2024; Corso et al., 2020; Garg et al., 2020; Sato et al., 2021). Most of these works analyzed GNNs in relation to the graph isomorphism task. Xu et al. (2018a) was among the first to lay the foundations for creating a maximally expressive GNN based on the Weisfeiler-Leman (WL) graph isomorphism test (Weisfeiler & Leman, 1968). Subsequent works build upon their results by considering extensions to the original 1-WL test. However, these results only hold with countable node feature representation which potentially limits their applicability. Meanwhile, Corso et al. (2020) proposed using multiple aggregators to create powerful GNNs when the space of node feature representation is uncountable. Interestingly, there has been no significant theoretical progress since this work.

This paper presents a novel perspective on the representational capability of GNNs when the space of node feature representation is uncountable. The key idea is to define a *pseudometric* distance on the space of input to create a *soft-injective* function such that distinct inputs may produce *similar* outputs if and only if the distance between the inputs is sufficiently small on some representation. This idea is investigated across all levels—node-level, neighborhood-level, and graph-level. From the theoretical results, a simple and computationally efficient *soft-isomorphic* relational graph convolution network (SIR-GCN) which emphasizes the contextualized transformation of neighborhood feature representations using *anisotropic* and *dynamic* message functions is proposed. The mathematical relationship between SIR-GCN and popular GNNs in literature is also presented to underscore the novelty and advantages of the proposed model. Experiments on synthetic and benchmark datasets in node and graph property prediction tasks then highlight the expressivity of SIR-GCN, positioning the proposed model as the best-performing MPNN instance.

## 2 GRAPH NEURAL NETWORKS

Let $\mathcal{G} = (\mathcal{V}_\mathcal{G}, \mathcal{E}_\mathcal{G})$ be a graph and $\mathcal{N}_\mathcal{G}(u) \subseteq \mathcal{V}_\mathcal{G}$ the set of nodes adjacent to node $u \in \mathcal{V}_\mathcal{G}$. The subscript $\mathcal{G}$ will be omitted whenever the context is clear. Suppose $\mathcal{H}$ is the space of node feature representation, henceforth feature, and $\boldsymbol{h_u} \in \mathcal{H}$ is the feature of node $u$. A GNN following the message-passing scheme can be expressed mathematically as

$$
\begin{aligned}
\boldsymbol{H_u} &= \{\!\{\boldsymbol{h_v} : v \in \mathcal{N}_\mathcal{G}(u)\}\!\} \\
\boldsymbol{a_u} &= \mathrm{AGG}\left(\boldsymbol{H_u}\right) \\
\boldsymbol{h_u^*} &= \mathrm{COMB}\left(\boldsymbol{h_u}, \boldsymbol{a_u}\right),
\end{aligned}
\tag{1}
$$

where AGG and COMB are some aggregation and combination strategies, respectively, $\boldsymbol{H_u}$ is the *multiset* (Xu et al., 2018a) of neighborhood features for node $u$, $\boldsymbol{a_u}$ is the aggregated neighborhood features for node $u$, and $\boldsymbol{h_u^*}$ is the updated feature for node $u$. Since AGG takes arbitrary-sized *multisets* of neighborhood features as input and transforms them into a single feature, it may be considered a hash function. Hence, aggregation and hash functions shall be used interchangeably throughout the paper.

**Related works**  When $\mathcal{H}$ is countable, Xu et al. (2018a) showed that there exists a function $f : \mathcal{H} \to \mathcal{S}$ such that the aggregation or hash function

$$
F\left(\boldsymbol{H}\right) = \sum_{\boldsymbol{h} \in \boldsymbol{H}} f\left(\boldsymbol{h}\right)
\tag{2}
$$

is injective or unique for each *multiset* of neighborhood features $\boldsymbol{H}$ of bounded size in the embedded feature space $\mathcal{S}$. This result forms the theoretical basis of GIN.

Meanwhile, the result above no longer holds when $\mathcal{H}$ is uncountable. In this setting, Corso et al. (2020) proved that if $\bigoplus$ comprises multiple aggregators (*e.g.*, mean, standard deviation, max, and min), the hash function

$$
M\left(\boldsymbol{H}\right) = \bigoplus_{\boldsymbol{h} \in \boldsymbol{H}} m\left(\boldsymbol{h}\right)
\tag{3}
$$

produces a unique output for every $\boldsymbol{H}$ of bounded size. This finding provides the foundation for the principal neighborhood aggregation (PNA) (Corso et al., 2020). Notably, for this result to hold, the number of aggregators in $\bigoplus$ must also scale with the size of the *multiset* of neighborhood features $\boldsymbol{H}$, which may be infeasible for large and dense graphs.

## 3 SOFT-INJECTIVE FUNCTIONS

While injective functions and metrics are necessary for tasks requiring strict isomorphism, many practical applications of GNNs often do not require such strict constraints. For instance, in node classification tasks, the model must produce identical outputs for some distinct nodes. Thus, this paper *softly* relaxes these constraints by employing *pseudometrics* and *soft-injective* functions.

**Definition 1** (Pseudometric). *Let $\mathcal{H}$ be a non-empty set. A function $d : \mathcal{H} \times \mathcal{H} \to \mathbb{R}_{\geq 0}$ is a pseudometric on $\mathcal{H}$ if the following holds for all $\boldsymbol{h}^{(1)}, \boldsymbol{h}^{(2)}, \boldsymbol{h}^{(3)} \in \mathcal{H}$:*

- $d\left(\boldsymbol{h}^{(1)}, \boldsymbol{h}^{(1)}\right) = 0$;

- $d\left(\boldsymbol{h}^{(1)}, \boldsymbol{h}^{(2)}\right) = d\left(\boldsymbol{h}^{(2)}, \boldsymbol{h}^{(1)}\right)$; and

- $d\left(\boldsymbol{h}^{(1)}, \boldsymbol{h}^{(3)}\right) \leq d\left(\boldsymbol{h}^{(1)}, \boldsymbol{h}^{(2)}\right) + d\left(\boldsymbol{h}^{(2)}, \boldsymbol{h}^{(3)}\right).$

Note that unlike a metric, $d\left(\boldsymbol{h}^{(1)}, \boldsymbol{h}^{(2)}\right) = 0 \;\not\Longrightarrow\; \boldsymbol{h}^{(1)} = \boldsymbol{h}^{(2)}$ for a *pseudometric $d$*. The following assumption is then imposed on the *psuedometric $d$*, leveraging results from kernel theory.

**Assumption 1.** *Let $d : \mathcal{H} \times \mathcal{H} \to \mathbb{R}_{\geq 0}$ be a pseudometric on $\mathcal{H}$ such that $-d^2$ is a conditionally positive definite kernel on $\mathcal{H}$.*

The Euclidean distance is an example of a *pseudometric* satisfying Assumption 1. A class of *pseudometrics* satisfying this assumption is provided below, see Berg et al. (1984) and Schölkopf (2000) for more.

**Remark 1.** *Consider the pseudometrics $d_1$ and $d_2$ on $\mathcal{H}$ satisfying Assumption 1. For $a > 0$ and $0 < p < 1$, the pseudometrics $a \cdot d_1$, $\sqrt{d_1^2 + d_2^2}$, and $d_1^p$ also satisfy Assumption 1.*

Assumption 1 thus offers considerable flexibility in the choice of *pseudometric $d$*. The following theorem then *softly* relaxes the injective and metric requirements in previous works.

**Theorem 1.** *Let $\mathcal{H}$ be a non-empty set with a pseudometric $d : \mathcal{H} \times \mathcal{H} \to \mathbb{R}_{\geq 0}$ satisfying Assumption 1. There exists a feature map $g : \mathcal{H} \to \mathcal{S}$ such that for every $\boldsymbol{h}^{(1)}, \boldsymbol{h}^{(2)} \in \mathcal{H}$ and $\varepsilon_1 > \varepsilon_2 > 0$,*

$$\varepsilon_2 < \left\| g\left(\boldsymbol{h}^{(1)}\right) - g\left(\boldsymbol{h}^{(2)}\right) \right\| < \varepsilon_1 \iff \varepsilon_2 < d\left(\boldsymbol{h}^{(1)}, \boldsymbol{h}^{(2)}\right) < \varepsilon_1. \tag{4}$$

$$d_u\left(\boldsymbol{h}_u^{(1)}, \boldsymbol{h}_u^{(2)}\right) < \varepsilon < d_u\left(\boldsymbol{h}_u^{(1)}, \boldsymbol{h}_u^{(3)}\right)$$

(a) Input feature space $\mathcal{H}$.

(b) Embedded feature space $\mathcal{S}$.

Figure 1: *Pseudometric $d_u$ and the corresponding feature map $g_u$.*

Theorem 1 shows that, for each node $u \in \mathcal{V}$, given a *pseudometric* distance $d_u$ that represents a *dissimilarity* function on $\mathcal{H}$, possibly encoded with prior knowledge, there exists a corresponding feature map $g_u$ that maps distinct inputs $\boldsymbol{h}_u^{(1)}, \boldsymbol{h}_u^{(2)} \in \mathcal{H}$ close in the embedded feature space $\mathcal{S}$ if and only if $d_u$ determines $\boldsymbol{h}_u^{(1)}, \boldsymbol{h}_u^{(2)}$ to be sufficiently *similar* on some representation. The lower bound $\varepsilon_2$ asserts the ability of $g_u$ to separate elements of $\mathcal{H}$ in the embedded feature space $\mathcal{S}$ while the upper bound $\varepsilon_1$ ensures $g_u$ maintains the relationship between elements of $\mathcal{H}$ with respect to $d_u$. The feature map $g_u$ may then be described as *soft-injective*.[1] Corollary 1 extends this result for *multisets*.

---

[1]The *pseudometric $d$* induces the equivalence class $[\boldsymbol{h}]_d := \{\boldsymbol{h}' \in \mathcal{H} : d(\boldsymbol{h}, \boldsymbol{h}') = 0\}$ with the quotient space $\mathcal{H}_d := \mathcal{H} \setminus d = \{[\boldsymbol{h}]_d : \boldsymbol{h} \in \mathcal{H}\}$ such that $d$ becomes metric and the corresponding feature map $g$ becomes injective on $\mathcal{H}_d$ (Schoenberg, 1938). Hence, $g$ may be described as *soft-injective*.

### 3.1 SOFT-ISOMORPHIC RELATIONAL GRAPH CONVOLUTION NETWORK

**Corollary 1.** *Let $\mathcal{H}$ be a non-empty set with a pseudometric $D$ on bounded, equinumerous multisets of $\mathcal{H}$ defined as*

$$D^2\left(\boldsymbol{H}^{(1)}, \boldsymbol{H}^{(2)}\right) = \sum_{\substack{\boldsymbol{h} \in \boldsymbol{H}^{(1)} \\ \boldsymbol{h}' \in \boldsymbol{H}^{(2)}}} d^2(\boldsymbol{h}, \boldsymbol{h}') - \frac{1}{2} \sum_{\substack{\boldsymbol{h} \in \boldsymbol{H}^{(1)} \\ \boldsymbol{h}' \in \boldsymbol{H}^{(1)}}} d^2(\boldsymbol{h}, \boldsymbol{h}') - \frac{1}{2} \sum_{\substack{\boldsymbol{h} \in \boldsymbol{H}^{(2)} \\ \boldsymbol{h}' \in \boldsymbol{H}^{(2)}}} d^2(\boldsymbol{h}, \boldsymbol{h}') \quad (5)$$

*for some pseudometric $d : \mathcal{H} \times \mathcal{H} \rightarrow \mathbb{R}_{\geq 0}$ satisfying Assumption 1 and bounded, equinumerous multisets $\boldsymbol{H}^{(1)}, \boldsymbol{H}^{(2)}$. There exists a feature map $g : \mathcal{H} \rightarrow \mathcal{S}$ such that for every $\boldsymbol{H}^{(1)}, \boldsymbol{H}^{(2)}$ and $\varepsilon_1 > \varepsilon_2 > 0$,*

$$\varepsilon_2 < \left\| G\left(\boldsymbol{H}^{(1)}\right) - G\left(\boldsymbol{H}^{(2)}\right) \right\| < \varepsilon_1 \iff \varepsilon_2 < D\left(\boldsymbol{H}^{(1)}, \boldsymbol{H}^{(2)}\right) < \varepsilon_1, \quad (6)$$

*where*

$$G(\boldsymbol{H}) = \sum_{\boldsymbol{h} \in \boldsymbol{H}} g(\boldsymbol{h}). \quad (7)$$

Similarly, Corollary 1 shows that, for each node $u \in \mathcal{V}$, given a *pseudometric* distance $D_u$ on *multisets* of $\mathcal{H}$ defined in Eqn. 5 with a corresponding *pseudometric* distance $d_u$ on $\mathcal{H}$, there exists a corresponding feature map $g_u$ and *soft-injective* hash function $G_u$ defined in Eqn. 7 that produces *similar* outputs for distinct *multisets* of neighborhood features $\boldsymbol{H}_u^{(1)}, \boldsymbol{H}_u^{(2)}$ if and only if $D_u$ deems $\boldsymbol{H}_u^{(1)}, \boldsymbol{H}_u^{(2)}$ to be sufficiently *similar* on some representation. Likewise, the lower and upper bounds guarantee the ability of $G_u$ to separate equinumerous *multisets* of $\mathcal{H}$ in the embedded feature space $\mathcal{S}$ while maintaining the relationship with respect to $D_u$. In this setting, the feature map $g_u$ may be interpreted as the message function (Gilmer et al., 2017) of the aggregation strategy that transforms the individual neighborhood features. Meanwhile, the *psuedometric* $D_u$ corresponds to the kernel distance (Joshi et al., 2011) which intuitively represents the difference between the cross-distance and self-distance between two *multisets*. The two necessary properties of the *soft-injective* message function—*dynamic* and *anisotropic*—are then motivated below.

***Dynamic* transformation** To illustrate the role of *pseudometrics*, consider node $u$ with two neighbors $v_1$ and $v_2$ and the task of anomaly detection on the scalar node features $\boldsymbol{h}_{v_1}$ and $\boldsymbol{h}_{v_2}$ representing zero-mean scores. If $d_u$ simply corresponds to the Euclidean distance, then the corresponding hash function $G_u$ becomes linear as presented in Fig. 2a. The contour plot highlights collisions—instances where distinct inputs produce identical outputs (*i.e.*, the equivalence class $[\boldsymbol{H}]_D$)—between *dissimilar multisets* of neighborhood features, resulting in aggregated neighborhood features that are less useful for the task.

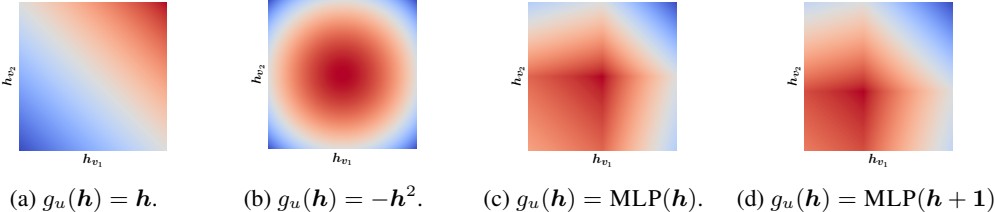

(a) $g_u(\boldsymbol{h}) = \boldsymbol{h}$.     (b) $g_u(\boldsymbol{h}) = -\boldsymbol{h}^2$.     (c) $g_u(\boldsymbol{h}) = \text{MLP}(\boldsymbol{h})$.     (d) $g_u(\boldsymbol{h}) = \text{MLP}(\boldsymbol{h} + \boldsymbol{1})$.

Figure 2: Hash functions $G_u$ under different message functions $g_u$.

Nevertheless, other choices of *pseudometrics*, possibly incorporating prior knowledge, would correspond to more complex message functions $g_u$. This leads to non-trivial hash functions $G_u$ and contour plots where only the regions determined by $D_u$ to be *similar* may produce *similar* aggregated neighborhood features, making collisions more informative and controlled. This also highlights the significance of *dynamic* (Brody et al., 2021) or non-linear message functions $g_u$ in MPNNs.

As further illustration, if $d_u$ instead corresponds to the Euclidean distance of the squared score, then the corresponding hash function $G_u$ has the contour plot in Fig. 2b. The resulting hash collisions and equivalence classes then become more useful and meaningful for detecting anomalous scores.

***Anisotropic* messages**   It is also worth noting that Corollary 1 holds for each node $u \in \mathcal{V}$ independently. Hence, different nodes may correspond to different $D_u$, $d_u$, $g_u$, and $G_u$. For simplicity, especially in inductive learning contexts, consider a single *pseudometric* instead, defined as

$$D^2 \left( \boldsymbol{H}_{\boldsymbol{u}}^{(1)}, \boldsymbol{H}_{\boldsymbol{u}}^{(2)}; \boldsymbol{h}_{\boldsymbol{u}} \right) = \sum_{\substack{\boldsymbol{h} \in \boldsymbol{H}_{\boldsymbol{u}}^{(1)} \\ \boldsymbol{h'} \in \boldsymbol{H}_{\boldsymbol{u}}^{(2)}}} d^2(\boldsymbol{h}, \boldsymbol{h'}; \boldsymbol{h}_{\boldsymbol{u}}) - \frac{1}{2} \sum_{\substack{\boldsymbol{h} \in \boldsymbol{H}_{\boldsymbol{u}}^{(1)} \\ \boldsymbol{h'} \in \boldsymbol{H}_{\boldsymbol{u}}^{(1)}}} d^2(\boldsymbol{h}, \boldsymbol{h'}; \boldsymbol{h}_{\boldsymbol{u}}) - \frac{1}{2} \sum_{\substack{\boldsymbol{h} \in \boldsymbol{H}_{\boldsymbol{u}}^{(2)} \\ \boldsymbol{h'} \in \boldsymbol{H}_{\boldsymbol{u}}^{(2)}}} d^2(\boldsymbol{h}, \boldsymbol{h'}; \boldsymbol{h}_{\boldsymbol{u}}),$$

(8)

with a single hash function, defined as

$$G \left( \boldsymbol{H}_{\boldsymbol{u}}; \boldsymbol{h}_{\boldsymbol{u}} \right) = \sum_{\boldsymbol{h} \in \boldsymbol{H}_{\boldsymbol{u}}} g \left( \boldsymbol{h}; \boldsymbol{h}_{\boldsymbol{u}} \right),$$

(9)

for every node $u \in \mathcal{V}$. This approach makes $D$, $d$, $g$, and $G$ *anisotropic* (Dwivedi et al., 2023) (*i.e.*, a function of both the features of the query (center) node $\boldsymbol{h}_{\boldsymbol{u}}$ and key (neighboring) nodes $\boldsymbol{h} \in \boldsymbol{H}_{\boldsymbol{u}}$). Moreover, contextualized on the features of the query node, $D$ may still be interpreted as a *pseudometric* controlling hash collisions with a corresponding *soft-injective* hash function $G$.

Furthermore, the integration of $\boldsymbol{h}_{\boldsymbol{u}}$ also allows for the interpretation of $g$ as a relational message function, guiding how features of the key nodes are to be embedded and transformed based on the features of the query node. Figs. 2c and 2d provide intuition for this idea where the introduction of a bias term, assuming a function of the features of the query node, shifts the contour plot to produce distinct aggregated neighborhood features $\boldsymbol{a}_{\boldsymbol{u}} \neq \boldsymbol{a}_{\boldsymbol{u'}}$ for nodes $u$ and $u'$ with identical neighborhood features $\boldsymbol{H}_{\boldsymbol{u}} = \boldsymbol{H}_{\boldsymbol{u'}}$ but distinct features $\boldsymbol{h}_{\boldsymbol{u}} \neq \boldsymbol{h}_{\boldsymbol{u'}}$. Nevertheless, one may also inject stochasticity into the node features to distinguish between nodes $u$ and $u'$ with identical features $\boldsymbol{h}_{\boldsymbol{u}} = \boldsymbol{h}_{\boldsymbol{u'}}$ and neighborhood features $\boldsymbol{H}_{\boldsymbol{u}} = \boldsymbol{H}_{\boldsymbol{u'}}$ with high probability (Sato et al., 2021) and to imitate having distinct $D_u$, $d_u$, $g_u$, and $G_u$ for each node $u \in \mathcal{V}$.

**Proposed model**   For a graph representation learning problem, one may directly model the *anisotropic* and *dynamic* relational message function $g$ as a two-layer multi-layer perceptron (MLP), with implicitly learned *pseudometrics*, following the universal approximation theorem (Hornik et al., 1989) to obtain the *soft-isomorphic* relational graph convolution network (SIR-GCN)

$$\boldsymbol{h}_{\boldsymbol{u}}^* = \sum_{v \in \mathcal{N}(u)} \boldsymbol{W}_{\boldsymbol{R}} \, \sigma \left( \boldsymbol{W}_{\boldsymbol{Q}} \boldsymbol{h}_{\boldsymbol{u}} + \boldsymbol{W}_{\boldsymbol{K}} \boldsymbol{h}_{\boldsymbol{v}} \right),$$

(10)

where $\sigma$ is a non-linear activation function, $\boldsymbol{W}_{\boldsymbol{Q}}, \boldsymbol{W}_{\boldsymbol{K}} \in \mathbb{R}^{d_{\text{hidden}} \times d_{\text{in}}}$, and $\boldsymbol{W}_{\boldsymbol{R}} \in \mathbb{R}^{d_{\text{out}} \times d_{\text{hidden}}}$. Leveraging linearity, the model has a computational complexity of

$$\mathcal{O} \left( |\mathcal{V}| \times d_{\text{hidden}} \times d_{\text{in}} + |\mathcal{E}| \times d_{\text{hidden}} + |\mathcal{V}| \times d_{\text{out}} \times d_{\text{hidden}} \right)$$

(11)

with computational efficiency achieved by the application of only an activation function along edges, making it comparable to conventional GNNs. Nevertheless, $\sigma$ may also be replaced with a deep MLP in practice if modeling $g$ as a shallow two-layer MLP becomes infeasible.

In essence, the proposed SIR-GCN is a simple, interpretable, and computationally efficient instance of the MPNN framework. Moreover, in contrast to other MPNN instances in literature, the proposed model emphasizes the *anisotropic* and *dynamic* transformation of neighborhood features to obtain contextualized messages.

## 3.2 Soft-Isomorphic Graph Readout Function

Corollary 1 also shows that, for each graph $\mathcal{G}$, given a *pseudometric* distance $d_{\mathcal{G}}$ on $\mathcal{H}$ with a corresponding *pseudometric* distance $D_{\mathcal{G}}$ on *multisets* of $\mathcal{H}$ defined in Eqn. 5, there exists a corresponding feature map $r_{\mathcal{G}}$ and *soft-injective* graph readout function $R_{\mathcal{G}}$ defined in Eqn. 7. While this result holds for each graph $\mathcal{G}$ independently, one may simply consider a single $D$, $d$, $r$, and $R$ for every graph $\{\mathcal{G}_d\}_{d \in \mathcal{D}}$ under task $\mathcal{D}$. Nevertheless, the graph context and structure may also be integrated into $D$, $d$, $r$, and $R$, through a virtual super node (Gilmer et al., 2017) for instance, to imitate having distinct $D_{\mathcal{G}}$, $d_{\mathcal{G}}$, $r_{\mathcal{G}}$, and $R_{\mathcal{G}}$ for each graph $\mathcal{G}$ and to further enhance its representational capability.

Similarly, for a graph representation learning problem, $r$ may also be directly modeled as an MLP, with implicitly learned *pseudometrics*, to obtain the *soft-isomorphic* graph readout function

$$\boldsymbol{h}_{\mathcal{G}} = \sum_{v \in \mathcal{V}_{\mathcal{G}}} \text{MLP}_R \left( \boldsymbol{h}_{\boldsymbol{v}} \right),$$

(12)

where $\text{MLP}_R$ corresponds to $r$ and $h_{\mathcal{G}}$ is the graph-level feature of graph $\mathcal{G}$.

# 4 MATHEMATICAL DISCUSSION

The mathematical relationship of SIR-GCN with GCN, GraphSAGE, GAT, GIN, and PNA are presented in this section to highlight the novelty and contribution. While activation functions and MLPs applied after each GNN layer play a significant role in the overall performance, the discussions only focus on the message-passing operation that defines GNNs. The relationship between SIR-GCN and the 1-WL test is also presented to contextualize the representational capability of the former.

## 4.1 GCN AND GRAPHSAGE

It may be shown that Corollary 1 holds up to a constant scale. Hence, the mean aggregation and symmetric mean aggregation, by extension, may be used in place of the sum aggregation. If one sets $\sigma$ as identity or $\text{PRELU}(\alpha = 1)$, $\boldsymbol{W_Q} = \boldsymbol{0}$, $\boldsymbol{W_R W_K} = \boldsymbol{W}$, and $\tilde{\mathcal{N}}(u) = \mathcal{N}(u) \cup \{u\}$, one obtains

$$h_u^* = \sum_{v \in \mathcal{N}(u)} \frac{1}{\sqrt{|\mathcal{N}(u)|}\sqrt{|\mathcal{N}(v)|}} \boldsymbol{W} h_v \tag{13}$$

and

$$h_u^* = \frac{1}{\left|\tilde{\mathcal{N}}(u)\right|} \sum_{v \in \tilde{\mathcal{N}}(u)} \boldsymbol{W} h_v \tag{14}$$

which recovers GCN and GraphSAGE with mean aggregation, respectively. Moreover, the sum aggregation may also be replaced with the max aggregation, albeit without theoretical justification, to recover GraphSAGE with max pooling. Thus, GCN and GraphSAGE may be viewed as instances of SIR-GCN.[2] The difference lies in the *isotropic* (Dwivedi et al., 2023) nature (*i.e.*, a function of only the features of the key nodes) of GCN and GraphSAGE and the use of non-linearities only in the combination strategy.

## 4.2 GAT

Moreover, in Brody et al. (2021), the attention mechanism of GATv2 is modeled as an MLP given by

$$e_{u,v} = \boldsymbol{a}_{\text{GAT}}^\top \text{LEAKYRELU}\left(\boldsymbol{W_{Q,\text{GAT}}} h_u + \boldsymbol{W_{K,\text{GAT}}} h_v\right), \tag{15}$$

with the message from node $v$ to node $u$ proportional to $\exp\left(e_{u,v}\right) \cdot \boldsymbol{W_{K,\text{GAT}}} h_v$. While the attention mechanism of GATv2 is *anisotropic* and *dynamic*, messages are nevertheless only linearly transformed with node $u$ only determining the degree of contribution through the scalar $e_{u,v}$. Meanwhile, SIR-GCN applies the concept of *anisotropic* and *dynamic* functions in Eqn. 15 to the message function, allowing the features of the query node to *dynamically* transform messages. Specifically, if $\boldsymbol{W_Q} = \boldsymbol{W_{Q,\text{GAT}}}$, $\boldsymbol{W_K} = \boldsymbol{W_{K,\text{GAT}}}$, $\sigma = \text{LEAKYRELU}$ and $\boldsymbol{W_R} = \boldsymbol{a}_{\text{GAT}}^\top$, one obtains

$$h_u^* = \sum_{v \in \mathcal{N}(u)} \boldsymbol{a}_{\text{GAT}}^\top \text{LEAKYRELU}\left(\boldsymbol{W_{Q,\text{GAT}}} h_u + \boldsymbol{W_{K,\text{GAT}}} h_v\right) \tag{16}$$

which shows Eqn. 15 becoming a contextualized message in the SIR-GCN model. Nevertheless, GAT and GATv2 may be recovered, up to a normalizing constant, with the appropriate parameters.

## 4.3 GIN

Likewise, within the proposed SIR-GCN model, one may explicitly add a residual connection in the combination strategy to obtain

$$h_u^* = \text{MLP}_{\text{Res}}(h_u) + \sum_{v \in \mathcal{N}(u)} \boldsymbol{W_R}\, \sigma\left(\boldsymbol{W_Q} h_u + \boldsymbol{W_K} h_v\right), \tag{17}$$

---

[2]GraphSAGE with LSTM aggregation is not included in this discussion.

where $\mathrm{MLP_{Res}}$ is a learnable residual network. If $\mathrm{MLP_{Res}}(\boldsymbol{h}) = (1 + \epsilon) \cdot \boldsymbol{h}$, $\sigma = \mathrm{PRELU}(\alpha = 1)$, $\boldsymbol{W_Q} = \boldsymbol{0}$, and $\boldsymbol{W_R}\boldsymbol{W_K} = \boldsymbol{I}$, then

$$\boldsymbol{h_u^*} = (1 + \epsilon) \cdot \boldsymbol{h_u} + \sum_{v \in \mathcal{N}(u)} \boldsymbol{h_v} \tag{18}$$

is equivalent to GIN. Hence, SIR-GCN with residual connection encompasses GIN.

### 4.4 PNA

Furthermore, while SIR-GCN and PNA approach the problem of uncountable node features differently, both models highlight the significance of *anisotropic* message functions considering both the features of the query and key nodes. The key difference lies with PNA using a *static* (Brody et al., 2021) or linear message function $m$ which translates to

$$m\left(\boldsymbol{h_v}, \boldsymbol{h_u}\right) = \boldsymbol{W_K}\boldsymbol{h_v} + \boldsymbol{W_Q}\boldsymbol{h_u} = \boldsymbol{W_K}\boldsymbol{h_v} + \boldsymbol{b_u}. \tag{19}$$

As a result, the influence of the query node on the aggregated neighborhood features is limited. For instance, when using mean, max, or min aggregators, the influence of the query node $u$ is restricted to the bias term $\boldsymbol{b_u}$. Moreover, with normalized moment aggregators, the bias term is effectively canceled out during the normalization process, further reducing the influence of the query node. Hence, PNA does not fully leverage its *anisotropic* nature, attributed to its heuristic application of multiple aggregators and scalers in a linear MPNN, thereby limiting its expressivity. In contrast, the *dynamic* nature of SIR-GCN allows for the non-linear embedding of the features of the query node $\boldsymbol{h_u}$ within the aggregated neighborhood features, thereby fully leveraging its *anisotropic* nature.

### 4.5 1-WL test

Additionally, in terms of graph isomorphism representational capability, SIR-GCN is comparable to a modified 1-WL test. Suppose $w_u^{(l)}$ is the WL node label of node $u$ at the $l$th WL-test iteration. The modified update equation is given by

$$w_u^{(l)} \leftarrow \mathrm{hash}\left(\left\{\!\!\left\{ \left[w_v^{(l-1)}, w_u^{(l-1)}\right] : v \in \mathcal{N}(u) \right\}\!\!\right\}\right), \tag{20}$$

where the modification lies in concatenating the label of the center node with every element of the *multiset* before hashing. This modification, while negligible when $\mathcal{H}$ is countable, becomes significant when $\mathcal{H}$ is uncountable as noted in the previous section. Thus, SIR-GCN inherits the theoretical capabilities (and limitations) of the 1-WL test.

### 4.6 SIR-GCN

Overall, SIR-GCN offers flexibility in two key dimensions of GNNs: aggregation strategy and message transformation. Consequently, it generalizes four prominent GNNs in literature—GCN, GraphSAGE, GAT, and GIN—ensuring that it is at least as expressive as these models. Notably, SIR-GCN sets itself apart from other GNNs as the first MPNN instance to incorporate both *anisotropic* and *dynamic* (*i.e.*, contextualized) messages within the MPNN framework, making it well-suited for heterophilous tasks (Bronstein et al., 2021) while remaining adaptable to homophilous tasks.

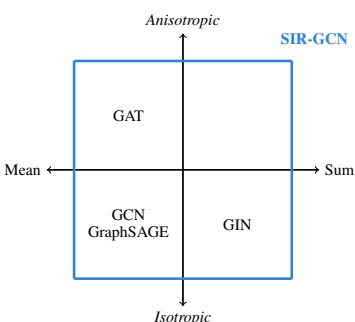

Figure 3: SIR-GCN encompasses GCN, GraphSAGE, GAT, and GIN.

In addition, SIR-GCN distinguishes itself from PNA by employing only a single aggregator that theoretically holds for graphs of arbitrary sizes, thus reducing computational complexity. Nevertheless, its expressivity is maintained through contextualized messages, allowing it to inherit the representational capability of the 1-WL test.

## 5 EXPERIMENTS

Experiments on synthetic and benchmark datasets in node and graph property prediction tasks are conducted to highlight the expressivity of SIR-GCN. To ensure fair evaluation, models not employing complex architectural design or manually crafted features using domain knowledge are used as primary comparisons.

### 5.1 SYNTHETIC DATASETS

**DictionaryLookup** DictionaryLookup (Brody et al., 2021) consists of bipartite graphs with $2n$ nodes—$n$ *key* nodes each with an attribute and value and $n$ *query* nodes each with an attribute. The task is to predict the value of *query* nodes by matching their attribute with the *key* nodes as in Fig. 4.

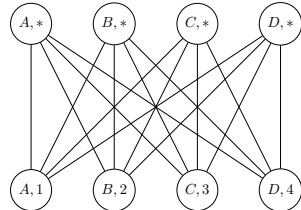

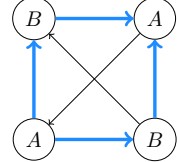

Figure 5: GraphHeterophily.

Figure 4: DictionaryLookup.

Table 1: Test accuracy on DictionaryLookup.

| Model | $n = 10$ | $n = 20$ | $n = 30$ | $n = 40$ | $n = 50$ |
|---|---|---|---|---|---|
| GCN | $0.10 \pm 0.00$ | $0.05 \pm 0.00$ | $0.03 \pm 0.00$ | $0.03 \pm 0.00$ | $0.02 \pm 0.00$ |
| GraphSAGE | $0.10 \pm 0.00$ | $0.05 \pm 0.00$ | $0.03 \pm 0.00$ | $0.02 \pm 0.00$ | $0.02 \pm 0.00$ |
| GATv2 | $0.99 \pm 0.03$ | $0.88 \pm 0.18$ | $0.74 \pm 0.28$ | $0.56 \pm 0.37$ | $0.60 \pm 0.40$ |
| GIN | $0.78 \pm 0.07$ | $0.29 \pm 0.03$ | $0.12 \pm 0.03$ | $0.03 \pm 0.00$ | $0.02 \pm 0.01$ |
| PNA | $1.00 \pm 0.00$ | $0.97 \pm 0.02$ | $0.86 \pm 0.09$ | $0.66 \pm 0.09$ | $0.50 \pm 0.05$ |
| **SIR-GCN** | $\mathbf{1.00 \pm 0.00}$ | $\mathbf{1.00 \pm 0.00}$ | $\mathbf{1.00 \pm 0.00}$ | $\mathbf{1.00 \pm 0.00}$ | $\mathbf{1.00 \pm 0.00}$ |

Table 1 presents the mean and standard deviation of the test accuracy for SIR-GCN, GCN, Graph-SAGE, GATv2, GIN, and PNA across different values of $n$. SIR-GCN and GATv2 achieve perfect accuracy attributed to their *anisotropic* and *dynamic* nature. However, it is observed that GATv2 suffers from performance degradation in some trials. Meanwhile, the other models fail to predict the value of *query* nodes even for the training graphs due to their *isotropic* and/or *static* nature. The results underscore the utility of a *dynamic* attentional or relational mechanism in capturing the relationship between the *query* and *key* nodes.

**GraphHeterophily** GraphHeterophily is an original synthetic dataset. It consists of random directed graphs with each node labeled one of $c$ classes. The task is then to count the total number of directed edges in each graph connecting nodes with distinct class labels as seen in Fig. 5.

Table 2: Test mean squared error on GraphHeterophily.

| Model | $c = 2$ | $c = 4$ | $c = 6$ | $c = 8$ | $c = 10$ |
|---|---|---|---|---|---|
| GCN | $22749 \pm 1242$ | $50807 \pm 2828$ | $62633 \pm 3491$ | $68965 \pm 3784$ | $72986 \pm 4025$ |
| GraphSAGE | $22962 \pm 1215$ | $36854 \pm 2330$ | $30552 \pm 1574$ | $21886 \pm 1896$ | $16529 \pm 1589$ |
| GATv2 | $22329 \pm 1307$ | $44972 \pm 2834$ | $49940 \pm 2942$ | $50063 \pm 3407$ | $49661 \pm 3488$ |
| GIN | $39.620 \pm 2.060$ | $37.193 \pm 1.382$ | $34.649 \pm 1.502$ | $32.424 \pm 1.841$ | $30.091 \pm 1.429$ |
| PNA | $172.15 \pm 97.82$ | $224.83 \pm 85.80$ | $249.99 \pm 108.56$ | $251.49 \pm 98.84$ | $195.72 \pm 36.65$ |
| **SIR-GCN** | $\mathbf{0.001 \pm 0.000}$ | $\mathbf{0.004 \pm 0.005}$ | $\mathbf{1.495 \pm 4.428}$ | $\mathbf{0.038 \pm 0.068}$ | $\mathbf{0.089 \pm 0.134}$ |

Table 2 presents the mean and standard deviation of the test mean squared error (MSE) for SIR-GCN, GCN, GraphSAGE, GATv2, GIN, and PNA across different values of $c$. SIR-GCN achieves near-zero MSE loss due to its *anisotropic* and *dynamic* nature and sum aggregation. In fact, if $\boldsymbol{W_Q} = \boldsymbol{I}$, $\boldsymbol{W_K} = -\boldsymbol{I}$, $\sigma = \text{RELU}$, and $\boldsymbol{W_R} = \boldsymbol{1}^\top$, SIR-GCN produces correct outputs for any graph. In

contrast, GCN, GraphSAGE, and GATv2 obtained large MSE losses due to their mean or max aggregation which fails to preserve the graph structure as noted by Xu et al. (2018a). Meanwhile, GIN and PNA successfully retain the graph structure but fail to learn the relationship between the labels of the query node and key nodes due to their *static* nature. The results illustrate the utility of *anisotropic* and *dynamic* models using sum aggregation even with countable node features.

## 5.2 BENCHMARK DATASETS

**Benchmarking GNNs**    Benchmarking GNNs (Dwivedi et al., 2023) is a collection of benchmark datasets consisting of diverse mathematical and real-world graphs across various GNN tasks. In particular, the WikiCS, PATTERN, and CLUSTER datasets fall under node property prediction tasks while the MNIST, CIFAR10, and ZINC datasets fall under graph property prediction tasks. Furthermore, the WikiCS, MNIST, and CIFAR10 datasets have uncountable node features while the remaining datasets have countable node features. The performance metric for ZINC is the mean absolute error (MAE) while the performance metric of the remaining datasets is accuracy. Dwivedi et al. (2023) provides more information regarding the individual datasets.

Table 3: Test performance on Benchmarking GNNs.

| Model | WikiCS (↑) | PATTERN (↑) | CLUSTER (↑) | MNIST (↑) | CIFAR10 (↑) | ZINC (↓) |
|---|---|---|---|---|---|---|
| MLP | $59.45 \pm 2.33$ | $50.52 \pm 0.00$ | $20.97 \pm 0.00$ | $95.34 \pm 0.14$ | $56.34 \pm 0.18$ | $0.706 \pm 0.006$ |
| GCN | $77.47 \pm 0.85$ | $85.50 \pm 0.05$ | $47.83 \pm 1.51$ | $90.12 \pm 0.15$ | $54.14 \pm 0.39$ | $0.416 \pm 0.006$ |
| GraphSAGE | $74.77 \pm 0.95$ | $50.52 \pm 0.00$ | $50.45 \pm 0.15$ | $97.31 \pm 0.10$ | $65.77 \pm 0.31$ | $0.468 \pm 0.003$ |
| GAT | $76.91 \pm 0.82$ | $75.82 \pm 1.82$ | $57.73 \pm 0.32$ | $95.54 \pm 0.21$ | $64.22 \pm 0.46$ | $0.475 \pm 0.007$ |
| GIN | $75.86 \pm 0.58$ | $85.59 \pm 0.01$ | $58.38 \pm 0.24$ | $96.49 \pm 0.25$ | $55.26 \pm 1.53$ | $0.387 \pm 0.015$ |
| GatedGCN | - | $84.48 \pm 0.12$ | $60.40 \pm 0.42$ | $97.34 \pm 0.14$ | $67.31 \pm 0.31$ | $0.435 \pm 0.011$ |
| PNA | - | - | - | $97.19 \pm 0.08$ | $70.21 \pm 0.15$ | $0.320 \pm 0.032$ |
| EGC-M | - | - | - | - | $71.03 \pm 0.42$ | $0.281 \pm 0.007$ |
| **SIR-GCN** | $\mathbf{78.06 \pm 0.66}$ | $\mathbf{85.75 \pm 0.03}$ | $\mathbf{63.35 \pm 0.19}$ | $\mathbf{97.90 \pm 0.08}$ | $\mathbf{71.98 \pm 0.40}$ | $\mathbf{0.278 \pm 0.024}$ |

Note: Missing values indicate that no results were published.

Table 3 presents the mean and standard deviation of the test performance for SIR-GCN and comparable GNN models across the six benchmarks where the experimental set-up follows that of Dwivedi et al. (2023) to ensure fair evaluation. The results show that SIR-GCN consistently outperforms popular GNNs in literature. Notably, SIR-GCN also outperforms both PNA (Corso et al., 2020) and efficient graph convolution (EGC-M) (Tailor et al., 2021) which use multiple aggregators. This highlights the significance of contextualized messages in enhancing the expressivity of GNNs, complementing the discussion in the previous section.

**ogbn-arxiv**    ogbn-arxiv (Hu et al., 2020) is a benchmark dataset representing the citation network between all Computer Science (CS) arXiv papers indexed by Microsoft academic graph (Wang et al., 2020). Each node represents an arXiv paper and a directed edge represents a citation. The task is to classify each paper, based on its title and abstract, into the 40 subject areas of arXiv CS papers.

Table 4: Test accuracy on ogbn-arxiv.

| Model | GIANT-XRT (Chien et al., 2021) | BoT (Wang et al., 2021) | C&S (Huang et al., 2020) | Others | Accuracy | Parameters |
|---|---|---|---|---|---|---|
| GATv2 | ✓ | | | | $0.7415 \pm 0.0005$ | 207,520 |
| GraphSAGE | ✓ | | | | $0.7435 \pm 0.0014$ | 546,344 |
| **SIR-GCN** | ✓ | | | | $0.7525 \pm 0.0009$ | 667,176 |
| | ✓ | ✓ | ✓ | | $\mathbf{0.7574 \pm 0.0020}$ | **697,896** |
| LGGNN | ✓ | ✓ | ✓ | | $0.7570 \pm 0.0018$ | 1,161,640 |
| RevGAT | ✓ | | | KD, DCN | $0.7636 \pm 0.0013$ | 1,304,912 |
| AGDN | ✓ | ✓ | | self-KD | $0.7637 \pm 0.0011$ | 1,309,760 |

Table 4 presents the mean and standard deviation of the test accuracy for SIR-GCN and other models in literature. The tricks used and the number of parameters are also presented for completeness. The results show that SIR-GCN, utilizing only a single GNN layer, outperforms comparable models in predicting the subject area of the papers. As expected, however, SIR-GCN fails to compete

with complex frameworks utilizing more tricks such as the reversible GAT (RevGAT) (Li et al., 2021) and the adaptive graph diffusion network (AGDN) (Sun et al., 2020), both of which build upon GAT by employing grouped reversible residual connections and adaptive graph diffusion, respectively. Nevertheless, SIR-GCN achieves performance close to that of the complex GNN frameworks mentioned, showcasing an effective balance between complexity and expressivity.

**ogbg-molhiv** ogbg-molhiv (Hu et al., 2020) is another benchmark dataset where each graph represents a molecule with nodes representing atoms and edges representing chemical bonds. Node features contain information regarding the atom while edge features contain information regarding the chemical bond. The task is to predict whether or not the molecules inhibit HIV replication.

Table 5: Test ROC-AUC on ogbg-molhiv.

| Model | GraphNorm (Cai et al., 2021) | VirtualNode (Gilmer et al., 2017) | Others | ROC-AUC | Parameters |
|---|---|---|---|---|---|
| GIN | | ✓ | FLAG | $0.7748 \pm 0.0096$ | 3,336,306 |
| GIN | ✓ | | | $0.7773 \pm 0.0129$ | 1,518,901 |
| EGC-M | | | | $0.7818 \pm 0.0153$ | 317,265 |
| GCN | ✓ | | | $0.7883 \pm 0.0100$ | 526,201 |
| PNA | | | | $0.7905 \pm 0.0132$ | 326,081 |
| **SIR-GCN** | | | | $0.7721 \pm 0.0110$ | 327,901 |
| | ✓ | | | $\mathbf{0.7981 \pm 0.0062}$ | **328,201** |
| GSN | | | | $0.7799 \pm 0.0100$ | 3,338,701 |
| GSAT | | | | $0.8067 \pm 0.0950$ | 249,602 |
| CIN | | | | $0.8094 \pm 0.0057$ | 239,745 |

Table 5 presents the mean and standard deviation of the test area under the receiver operating characteristic curve (ROC-AUC) for SIR-GCN and other models in literature. The tricks used and the number of parameters are also presented for completeness. The results show that with only a single GNN layer, SIR-GCN outperforms comparable models in predicting molecules inhibiting HIV replication, highlighting its expressivity. Given the simplicity of SIR-GCN, it is expected to exhibit lower performance compared to complex models such as the graph stochastic attention (GSAT) (Miao et al., 2022), which builds upon PNA by leveraging the information bottleneck principle, and the cell isomorphism network (CIN) (Bodnar et al., 2021), which is a hierarchical message-passing framework utilizing the topological features of graphs. Despite its simple design, the expressivity of SIR-GCN is evident in its close performance to that of complex GNN frameworks.

## 6 CONCLUSION

In summary, the paper provides a novel perspective for creating a powerful GNN across all levels when the space of node features is uncountable. The central idea is to use *pseudometric* distances to create *soft-injective* functions such that distinct inputs may produce *similar* outputs if and only if the distance between inputs is sufficiently small on some representation. From the results, the SIR-GCN is proposed as the first MPNN instance to emphasize contextualized message transformation, setting it apart from other GNNs. This design also enables it to learn the complex relationships between neighboring nodes and allows it to better handle uncountable node features. Furthermore, the model is shown to generalize classical GNN methodologies. Despite its simple design, empirical results underscore SIR-GCN as the best-performing MPNN instance that effectively balances model complexity and expressivity. The paper thus contributes to GNN literature by theoretically and empirically demonstrating the necessity of both *anisotropic* and *dynamic* messages to enhance GNN expressivity. Future works may consider incorporating SIR-GCN into complex frameworks, such as grouped reversible residual connections (Li et al., 2021), adaptive graph diffusion (Sun et al., 2020), graph stochastic attention (Miao et al., 2022), and hierarchical message-passing (Bodnar et al., 2021), to address the limitations inherent in the MPNN framework and to develop more expressive GNNs.

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

## A  Proofs

**Definition 2** (Conditionally positive definite kernel (Schölkopf, 2000)). *Let $\mathcal{H}$ be a non-empty set. A symmetric function $\tilde{k} : \mathcal{H} \times \mathcal{H} \to \mathbb{R}$ is a conditionally positive definite kernel on $\mathcal{H}$ if for all $N \in \mathbb{N}$ and $\boldsymbol{h}^{(1)}, \boldsymbol{h}^{(2)}, \ldots, \boldsymbol{h}^{(N)} \in \mathcal{H}$,*

$$\sum_{i=1}^{N} \sum_{j=1}^{N} c_i c_j \, \tilde{k}\left(\boldsymbol{h}^{(i)}, \boldsymbol{h}^{(j)}\right) \geq 0, \tag{21}$$

*with $c_1, c_2, \ldots, c_N \in \mathbb{R}$ and $\sum_{i=1}^{N} c_i = 0$.*

**Theorem 2** (Hilbert space representation of conditionally positive definite kernels (Berg et al., 1984; Schoenberg, 1938; Schölkopf, 2000)). *Let $\mathcal{H}$ be a non-empty set and $\tilde{k} : \mathcal{H} \times \mathcal{H} \to \mathbb{R}$ a conditionally positive definite kernel on $\mathcal{H}$ satisfying $\tilde{k}(\boldsymbol{h}, \boldsymbol{h}) = 0$ for all $\boldsymbol{h} \in \mathcal{H}$. There exists a Hilbert space $\mathcal{S}$ of real-valued functions on $\mathcal{H}$ and a feature map $g : \mathcal{H} \to \mathcal{S}$ such that for every $\boldsymbol{h}^{(1)}, \boldsymbol{h}^{(1)} \in \mathcal{H}$,*

$$\left\| g\left(\boldsymbol{h}^{(1)}\right) - g\left(\boldsymbol{h}^{(2)}\right) \right\|^2 = -\tilde{k}\left(\boldsymbol{h}^{(1)}, \boldsymbol{h}^{(2)}\right). \tag{22}$$

*Proof.* See Schölkopf (2000). □

**Theorem 1.** *Let $\mathcal{H}$ be a non-empty set with a pseudometric $d : \mathcal{H} \times \mathcal{H} \to \mathbb{R}_{\geq 0}$ satisfying Assumption 1. There exists a feature map $g : \mathcal{H} \to \mathcal{S}$ such that for every $\boldsymbol{h}^{(1)}, \boldsymbol{h}^{(2)} \in \mathcal{H}$ and $\varepsilon_1 > \varepsilon_2 > 0$,*

$$\varepsilon_2 < \left\| g\left(\boldsymbol{h}^{(1)}\right) - g\left(\boldsymbol{h}^{(2)}\right) \right\| < \varepsilon_1 \iff \varepsilon_2 < d\left(\boldsymbol{h}^{(1)}, \boldsymbol{h}^{(2)}\right) < \varepsilon_1. \tag{4}$$

*Proof.* Let $d : \mathcal{H} \times \mathcal{H} \to \mathbb{R}_{\geq 0}$ be a *pseudometric*. From Assumption 1 and Theorem 2, there exists a feature map $g : \mathcal{H} \to \mathcal{S}$ such that for every $\boldsymbol{h}^{(1)}, \boldsymbol{h}^{(2)} \in \mathcal{H}$,

$$\left\| g\left(\boldsymbol{h}^{(1)}\right) - g\left(\boldsymbol{h}^{(2)}\right) \right\| = d\left(\boldsymbol{h}^{(1)}, \boldsymbol{h}^{(2)}\right). \tag{23}$$

Hence, for every $\varepsilon_1 > \varepsilon_2 > 0$,

$$\varepsilon_2 < \left\| g\left(\boldsymbol{h}^{(1)}\right) - g\left(\boldsymbol{h}^{(2)}\right) \right\| < \varepsilon_1 \iff \varepsilon_2 < d\left(\boldsymbol{h}^{(1)}, \boldsymbol{h}^{(2)}\right) < \varepsilon_1. \tag{24}$$

$\square$

**Theorem 3.** *Suppose $\boldsymbol{h}^{(0)}, \boldsymbol{h}^{(1)}, \boldsymbol{h}^{(2)} \in \mathcal{H}$ and $\tilde{k} : \mathcal{H} \times \mathcal{H} \to \mathbb{R}$ is a symmetric function. Then*

$$k\left(\boldsymbol{h}^{(1)}, \boldsymbol{h}^{(2)}\right) = \frac{1}{2}\left[\tilde{k}\left(\boldsymbol{h}^{(1)}, \boldsymbol{h}^{(2)}\right) - \tilde{k}\left(\boldsymbol{h}^{(1)}, \boldsymbol{h}^{(0)}\right) - \tilde{k}\left(\boldsymbol{h}^{(0)}, \boldsymbol{h}^{(2)}\right) + \tilde{k}\left(\boldsymbol{h}^{(0)}, \boldsymbol{h}^{(0)}\right)\right] \tag{25}$$

*is positive definite if and only if $\tilde{k}$ is conditionally positive definite.*

*Proof.* See Schölkopf (2000). $\square$

**Corollary 1.** *Let $\mathcal{H}$ be a non-empty set with a pseudometric $D$ on bounded, equinumerous multisets of $\mathcal{H}$ defined as*

$$D^2\left(\boldsymbol{H}^{(1)}, \boldsymbol{H}^{(2)}\right) = \sum_{\substack{\boldsymbol{h} \in \boldsymbol{H}^{(1)} \\ \boldsymbol{h}' \in \boldsymbol{H}^{(2)}}} d^2(\boldsymbol{h}, \boldsymbol{h}') - \frac{1}{2}\sum_{\substack{\boldsymbol{h} \in \boldsymbol{H}^{(1)} \\ \boldsymbol{h}' \in \boldsymbol{H}^{(1)}}} d^2(\boldsymbol{h}, \boldsymbol{h}') - \frac{1}{2}\sum_{\substack{\boldsymbol{h} \in \boldsymbol{H}^{(2)} \\ \boldsymbol{h}' \in \boldsymbol{H}^{(2)}}} d^2(\boldsymbol{h}, \boldsymbol{h}') \tag{5}$$

*for some pseudometric $d : \mathcal{H} \times \mathcal{H} \to \mathbb{R}_{\geq 0}$ satisfying Assumption 1 and bounded, equinumerous multisets $\boldsymbol{H}^{(1)}, \boldsymbol{H}^{(2)}$. There exists a feature map $g : \mathcal{H} \to \mathcal{S}$ such that for every $\boldsymbol{H}^{(1)}, \boldsymbol{H}^{(2)}$ and $\varepsilon_1 > \varepsilon_2 > 0$,*

$$\varepsilon_2 < \left\| G\left(\boldsymbol{H}^{(1)}\right) - G\left(\boldsymbol{H}^{(2)}\right) \right\| < \varepsilon_1 \iff \varepsilon_2 < D\left(\boldsymbol{H}^{(1)}, \boldsymbol{H}^{(2)}\right) < \varepsilon_1, \tag{6}$$

*where*

$$G(\boldsymbol{H}) = \sum_{\boldsymbol{h} \in \boldsymbol{H}} g(\boldsymbol{h}). \tag{7}$$

*Proof.* Let $D$ be a *pseudometric* on bounded, equinumerous *multisets* of $\mathcal{H}$ defined as

$$D^2\left(\boldsymbol{H}^{(1)}, \boldsymbol{H}^{(2)}\right) = \sum_{\substack{\boldsymbol{h} \in \boldsymbol{H}^{(1)} \\ \boldsymbol{h}' \in \boldsymbol{H}^{(2)}}} d^2(\boldsymbol{h}, \boldsymbol{h}') - \frac{1}{2}\sum_{\substack{\boldsymbol{h} \in \boldsymbol{H}^{(1)} \\ \boldsymbol{h}' \in \boldsymbol{H}^{(1)}}} d^2(\boldsymbol{h}, \boldsymbol{h}') - \frac{1}{2}\sum_{\substack{\boldsymbol{h} \in \boldsymbol{H}^{(2)} \\ \boldsymbol{h}' \in \boldsymbol{H}^{(2)}}} d^2(\boldsymbol{h}, \boldsymbol{h}') \tag{26}$$

for some *pseudometric* $d : \mathcal{H} \times \mathcal{H} \to \mathbb{R}_{\geq 0}$ and bounded, equinumerous *multisets* $\boldsymbol{H}^{(1)}, \boldsymbol{H}^{(2)}$. From Assumption 1 and Theorem 3, the *pseudometric* $d$ has a corresponding positive definite kernel $k : \mathcal{H} \times \mathcal{H} \to \mathbb{R}$. A simple algebraic manipulation and using the fact that $\boldsymbol{H}^{(1)}, \boldsymbol{H}^{(2)}$ are equinumerous results in

$$D^2\left(\boldsymbol{H}^{(1)}, \boldsymbol{H}^{(2)}\right) = \sum_{\substack{\boldsymbol{h} \in \boldsymbol{H}^{(1)} \\ \boldsymbol{h}' \in \boldsymbol{H}^{(1)}}} k(\boldsymbol{h}, \boldsymbol{h}') + \sum_{\substack{\boldsymbol{h} \in \boldsymbol{H}^{(2)} \\ \boldsymbol{h}' \in \boldsymbol{H}^{(2)}}} k(\boldsymbol{h}, \boldsymbol{h}') - 2\sum_{\substack{\boldsymbol{h} \in \boldsymbol{H}^{(1)} \\ \boldsymbol{h}' \in \boldsymbol{H}^{(2)}}} k(\boldsymbol{h}, \boldsymbol{h}'). \tag{27}$$

Note that $D$ is indeed a *pseudometric* since $k$ is positive definite as noted by Joshi et al. (2011).[3] By the reproducing property of $k$ and the linearity of the inner product, it may be shown that

$$\left\| G\left(\boldsymbol{H}^{(1)}\right) - G\left(\boldsymbol{H}^{(2)}\right) \right\| = D\left(\boldsymbol{H}^{(1)}, \boldsymbol{H}^{(2)}\right), \tag{28}$$

where

$$G(\boldsymbol{H}) = \sum_{\boldsymbol{h} \in \boldsymbol{H}} g(\boldsymbol{h}) \tag{29}$$

and $g$ is the corresponding feature map of the kernel $k$. Hence, for every $\varepsilon_1 > \varepsilon_2 > 0$,

$$\varepsilon_2 < \left\| G\left(\boldsymbol{H}^{(1)}\right) - G\left(\boldsymbol{H}^{(2)}\right) \right\| < \varepsilon_1 \iff \varepsilon_2 < D\left(\boldsymbol{H}^{(1)}, \boldsymbol{H}^{(2)}\right) < \varepsilon_1. \tag{30}$$

$\square$

---

[3]If $k$ is also *integrally strictly positive definite* (Sriperumbudur et al., 2010), then the hash function $G$ becomes injective and $D$ becomes a metric.

# B    EXPERIMENTAL SET-UP

All experiments are conducted on a single NVIDIA® Quadro RTX 6000 (24GB) card using the Deep Graph Library (DGL) (Wang et al., 2019a) with PyTorch (Paszke et al., 2019) backend. For synthetic datasets, the reported results are obtained from the models at the final epoch across 10 trials with varying seed values. For benchmark datasets, the reported results are obtained from the models with the best validation loss across the 10 trials. The hyperparameters are chosen based on previous results and heuristics without extensive tuning.

## B.1    SYNTHETIC DATASETS

**DictionaryLookup**    Adopting Brody et al. (2021), the training dataset consists of 4,000 bipartite graphs, each containing $2n$ nodes with randomly assigned attributes and values, while the test dataset comprises 1,000 bipartite graphs with the same configuration. All models utilize a single GNN layer with $4n$ hidden units. A two-layer MLP is also used for GIN and $\sigma$ of SIR-GCN while PNA uses the sum, max, and std aggregators. Model training is performed with the AdamW (Loshchilov & Hutter, 2017) optimizer for over 500 epochs with a batch size of 256 and a learning rate of 0.001 that decays by a factor of 0.5 with patience of 10 epochs based on the training loss.

**GraphHeterophily**    The training dataset consists of 4,000 directed graphs, each containing a maximum of 50 nodes with uniformly selected edges using the `rand_graph` function of DGL and uniformly assigned node labels from one of $c$ classes using the `randint` function of PyTorch. These measures ensure that the graphs are sufficiently diverse with respect to graph structure and heterophily. Meanwhile, the test dataset comprises 1,000 directed graphs with the same configuration. All models utilize a single GNN layer with $10c$ hidden units and sum pooling as the graph readout function. A feed-forward neural network is also used for GIN while PNA uses the sum, max, and std aggregators. Model training is performed with the AdamW (Loshchilov & Hutter, 2017) optimizer for over 500 epochs with a batch size of 256 and a learning rate of 0.001 that decays by a factor of 0.5 with patience of 10 epochs based on the training loss.

## B.2    BENCHMARK DATASETS

**Benchmarking GNNs**    The datasets are obtained from `dgl` with data splits (training, validation, test) following Dwivedi et al. (2023). In line with Dwivedi et al. (2023), all models utilize 4 GNN layers with batch normalization and residual connections while constrained with a parameter budget of 100,000. Regularization with weights in $\left\{1 \times 10^{-7}, 1 \times 10^{-6}, 1 \times 10^{-5}\right\}$ and dropouts with rates in $\{0.1, 0.2, 0.3\}$ are also used to prevent overfitting. The mean, symmetric mean, and max aggregators are used since the sum aggregator is observed to not generalize well to unseen graphs as noted by Veličković et al. (2019). Additionally, sum pooling is used as the graph readout function for ZINC while mean pooling is used for MNIST and CIFAR10. Model training is performed with the AdamW (Loshchilov & Hutter, 2017) optimizer for over a maximum of 500 epochs with a batch size of 128, whenever applicable, and a learning rate of 0.001 that decays by a factor of 0.5 with patience of 10 epochs based on the training loss. The reported results for other models in Table 3 are obtained from Dwivedi et al. (2023), Corso et al. (2020), and Tailor et al. (2021).

**ogbn-arxiv**    The dataset is obtained from `ogb` with data splits (training, validation, test) following Hu et al. (2020). Furthermore, the GIANT-XRT (Chien et al., 2021) node features are also used, resulting in 768-dimensional input node features. The models utilize a single GNN layer with 256 hidden units, batch normalization, and residual connections. Regularization with weight $1 \times 10^{-6}$ and dropouts with rates in increments of 0.1 are also used to prevent overfitting. The symmetric mean aggregator is used along with existing tricks in literature. Model training is performed with the AdamW (Loshchilov & Hutter, 2017) optimizer for over 500 epochs and a learning rate of 0.01 that decays by a factor of 0.5 with patience of 50 epochs based on the training loss. The reported results for other models in Table 4 are obtained from the OGB leaderboard accessible at `https://ogb.stanford.edu`.

**ogbg-molhiv**    The dataset is obtained from `ogb` with data splits (training, validation, test) following Hu et al. (2020) and 174-dimensional input node feature embeddings. The models utilize a single

GNN layer, modified to leverage edge features as described in Appendix E, with 300 hidden units, batch/graph normalization, and residual connections. Regularization with weight $1 \times 10^{-7}$ and dropouts with rates in $\{0.1, 0.4\}$ are also used to prevent overfitting. The sum aggregator is used for SIR-GCN aggregation while mean pooling is used as the graph readout function. Model training is performed with the AdamW (Loshchilov & Hutter, 2017) optimizer for over 200 epochs with a batch size of 128 and a learning rate of 0.001 that decays by a factor of 0.5 with patience of 20 epochs based on the training loss. The reported results for other models in Table 5 are obtained from the OGB leaderboard accessible at `https://ogb.stanford.edu`.

## C  RUNTIME ANALYSIS

As an additional evaluation, the validation runtime for each model in the synthetic datasets is presented in Tables 6 and 7. The results, when considered alongside Tables 1 and 2, illustrate that SIR-GCN achieves a balance between computational complexity and model expressivity, specifically with regards to PNA which is also designed for uncountable node features but requires significantly longer runtime. Table 8 complements these results and further highlights how SIR-GCN has a computational runtime complexity comparable to GCN, GraphSAGE, GAT, GATv2, and GIN while outperforming these models across all benchmarks. Notably, SIR-GCN also demonstrates a lower complexity than PNA, yet delivers superior performance across all datasets. These additional analyses further underscore the practical utility of the proposed model.

Table 6: DictionaryLookup validation runtime.

| Model | $n = 10$ | $n = 20$ | $n = 30$ | $n = 40$ | $n = 50$ |
|---|---|---|---|---|---|
| GCN | $0.3526s \pm 0.0778s$ | $0.4734s \pm 0.0468s$ | $0.4777s \pm 0.0854s$ | $0.5619s \pm 0.0518s$ | $0.5520s \pm 0.0679s$ |
| GraphSAGE | $0.4565s \pm 0.0873s$ | $0.5264s \pm 0.0317s$ | $0.5716s \pm 0.1132s$ | $0.7742s \pm 0.0597s$ | $0.9193s \pm 0.0473s$ |
| GATv2 | $0.3950s \pm 0.1017s$ | $0.5276s \pm 0.0556s$ | $0.6191s \pm 0.0879s$ | $0.7472s \pm 0.0346s$ | $1.0065s \pm 0.0280s$ |
| GIN | $0.3696s \pm 0.0899s$ | $0.4610s \pm 0.0459s$ | $0.4670s \pm 0.0781s$ | $0.5947s \pm 0.0548s$ | $0.5194s \pm 0.0993s$ |
| PNA | $0.8854s \pm 0.0412s$ | $1.1913s \pm 0.1024s$ | $1.4526s \pm 0.0684s$ | $1.8793s \pm 0.0528s$ | $2.8387s \pm 0.0603s$ |
| SIR-GCN | $0.4687s \pm 0.0777s$ | $0.6066s \pm 0.0398s$ | $0.8053s \pm 0.0485s$ | $1.1496s \pm 0.0427s$ | $1.7031s \pm 0.0458s$ |

Table 7: GraphHeterophily validation runtime.

| Model | $c = 2$ | $c = 4$ | $c = 6$ | $c = 8$ | $c = 10$ |
|---|---|---|---|---|---|
| GCN | $0.4243s \pm 0.0520s$ | $0.3852s \pm 0.0517s$ | $0.3868s \pm 0.0743s$ | $0.4166s \pm 0.0551s$ | $0.4177s \pm 0.0494s$ |
| GraphSAGE | $0.4691s \pm 0.0400s$ | $0.4790s \pm 0.0440s$ | $0.4399s \pm 0.0629s$ | $0.4501s \pm 0.0603s$ | $0.4964s \pm 0.0601s$ |
| GATv2 | $0.4710s \pm 0.0978s$ | $0.4941s \pm 0.0567s$ | $0.4718s \pm 0.0361s$ | $0.5514s \pm 0.0608s$ | $0.5437s \pm 0.0724s$ |
| GIN | $0.4085s \pm 0.0741s$ | $0.3875s \pm 0.0627s$ | $0.3855s \pm 0.0645s$ | $0.4298s \pm 0.0566s$ | $0.4329s \pm 0.0534s$ |
| PNA | $2.2963s \pm 0.0413s$ | $2.4238s \pm 0.0611s$ | $2.4577s \pm 0.0533s$ | $2.4741s \pm 0.0665s$ | $2.5623s \pm 0.0425s$ |
| SIR-GCN | $0.5338s \pm 0.0353s$ | $0.5264s \pm 0.0737s$ | $0.5635s \pm 0.0695s$ | $0.5764s \pm 0.0401s$ | $0.6230s \pm 0.0388s$ |

Table 8: Asymptotic runtime complexity.

| Model | Complexity |
|---|---|
| GCN | $\mathcal{O}\left(|\mathcal{V}| \times d_{\text{out}} \times d_{\text{in}} + |\mathcal{E}| \times d_{\text{out}}\right)$ |
| GraphSAGE | $\mathcal{O}\left(|\mathcal{V}| \times d_{\text{out}} \times d_{\text{in}} + |\mathcal{E}| \times d_{\text{out}}\right)$ |
| GAT / GATv2 | $\mathcal{O}\left(|\mathcal{V}| \times d_{\text{out}} \times d_{\text{in}} + |\mathcal{E}| \times d_{\text{out}}\right)$ |
| GIN | $\mathcal{O}\left(|\mathcal{E}| \times d_{\text{in}} + |\mathcal{V}| \times \text{MLP}\right)$ |
| PNA | $\mathcal{O}\left(|\mathcal{E}| \times d_{\text{in}}^2 + |\mathcal{E}| \times d_{\text{in}} \times k + |\mathcal{V}| \times d_{\text{out}} \times d_{\text{in}} \times k\right)$ |
| SIR-GCN | $\mathcal{O}\left(|\mathcal{V}| \times d_{\text{hidden}} \times d_{\text{in}} + |\mathcal{E}| \times d_{\text{hidden}} + |\mathcal{V}| \times d_{\text{out}} \times d_{\text{hidden}}\right)$ |

Note: $k$ represents the number of aggregators and scalers in PNA.

## D  ADDITIONAL EXPERIMENTS

Additional experiments are conducted to further highlight the utility and novelty of SIR-GCN as the first MPNN instance to theoretically and empirically justify the use of *anisotropic* and *dynamic* message functions. Specifically, consider SIR-GCN (*static*), which uses linear messages by setting $\sigma$ as identity and $\boldsymbol{W_R} = \boldsymbol{I}$, and SIR-GCN (*isotropic*), which removes the dependency of messages

on the query node features $h_u$ by setting $W_Q = 0$. Table 9 presents the results for the SIR-GCN variants on the Benchmarking GNNs datasets. Although SIR-GCN achieves lower accuracy on WikiCS compared to the two simpler SIR-GCNs (*static* and *isotropic*), this result is consistent with the characteristics of the dataset. As noted by Dwivedi et al. (2023), WikiCS is a single-graph dataset with denser node neighborhoods and shorter average path lengths, which can make more expressive models like SIR-GCN prone to overfitting and oversmoothing. Thus, the simpler SIR-GCNs are naturally less expressive and achieve higher accuracies for this small dataset. In contrast, on larger and more complex datasets such as PATTERN, CLUSTER, MNIST, CIFAR10, and ZINC, SIR-GCN consistently outperforms both the simpler SIR-GCNs and conventional GNNs. This underscores the strong utility of both *anisotropic* and *dynamic* message functions in improving GNN representational capability. Overall, these additional results highlight the novelty of SIR-GCN and further confirm the theoretical and practical contributions of the paper in advancing GNN research.

Table 9: Additional experiments on Benchmarking GNNs.

| Model | WikiCS ($\uparrow$) | PATTERN ($\uparrow$) | CLUSTER ($\uparrow$) | MNIST ($\uparrow$) | CIFAR10 ($\uparrow$) | ZINC ($\downarrow$) |
|---|---|---|---|---|---|---|
| SIR-GCN (*static*) | $78.52 \pm 0.57$ | $85.72 \pm 0.02$ | $61.90 \pm 0.25$ | $95.65 \pm 0.84$ | $50.09 \pm 3.20$ | $0.334 \pm 0.014$ |
| SIR-GCN (*isotropic*) | $\mathbf{78.73 \pm 0.63}$ | $85.74 \pm 0.03$ | $62.60 \pm 0.38$ | $97.44 \pm 0.11$ | $68.88 \pm 0.27$ | $0.281 \pm 0.024$ |
| SIR-GCN | $78.06 \pm 0.66$ | $\mathbf{85.75 \pm 0.03}$ | $\mathbf{63.35 \pm 0.19}$ | $\mathbf{97.90 \pm 0.08}$ | $\mathbf{71.98 \pm 0.40}$ | $\mathbf{0.278 \pm 0.024}$ |

## E  SIR-GCN EXTENSIONS

Denote $h_{u,v}$ as the feature of the edge connecting node $v$ to node $u$. Following the intuition presented in Eqns. 8 and 9, SIR-GCN with residual connection may be modified to leverage edge features to obtain

$$h_u^* = \mathrm{MLP}_{\mathrm{Res}}(h_u) + \sum_{v \in \mathcal{N}(u)} W_R \, \sigma \left( W_Q h_u + W_E h_{u,v} + W_K h_v \right), \tag{31}$$

where $W_E \in \mathbb{R}^{d_{\mathrm{hidden}} \times d_{\mathrm{in}}}$. Consequently, this also increases the computational complexity of the model to

$$\mathcal{O} \left( |\mathcal{E}| \times d_{\mathrm{hidden}} \times d_{\mathrm{in}} + |\mathcal{V}| \times d_{\mathrm{out}} \times d_{\mathrm{hidden}} + |\mathcal{V}| \times \mathrm{MLP}_{\mathrm{Res}} \right), \tag{32}$$

with $\mathrm{MLP}_{\mathrm{Res}}$ denoting the computational complexity of $\mathrm{MLP}_{\mathrm{Res}}$, which is comparable to PNA. Similarly, this extension may be viewed as a generalization of GIN with edge features (Hu et al., 2019).

Furthermore, one may inject inductive bias into the *pseudometrics* which may correspond to specifying the architecture type for the corresponding message function $g$. For instance, if node features are known to have a sequential relationship (*e.g.*, stock (Hsu et al., 2021) and fMRI (Kim & Ye, 2020) data), $g$ may then be aptly modeled using recurrent or convolutional networks.

