# OpenReview forum: "Contextualized Messages Boost Graph Representations"
_ICLR.cc/2025/Conference — ICLR 2025 Conference Withdrawn Submission_

### Official Review · Reviewer_YvUg · 2024-11-02

**Soundness:** 3
**Presentation:** 3
**Contribution:** 3
**Rating:** 6
**Confidence:** 3

**Summary:**

The paper introduces a new perspective on the representational capability of GNNs by presenting a soft-injective function using a pseudometric distance. It then proposes a new message-passing scheme that performs competitively across various datasets.

**Strengths:**

1. The paper presents an interesting approach, offering clear insights into relaxing the injective constraint in message-passing processes by defining a pseudometric distance, which captures differences within data effectively.

2. The theoretical basis is solid, providing a coherent and accessible framework that helps explain the introduced ideas.

3. Although the proposed method is limited by the 1-WL test, it shows strong representational capabilities. Its effectiveness is supported by experiments conducted on both synthetic and real-world datasets, confirming its practical usefulness.

**Weaknesses:**

1. The experimental section is somewhat unclear, especially regarding the number of parameters in Table 4 compared to Table 3. It is confusing why the same model appears to have twice the number of parameters. The authors emphasize "a single layer" in line 507 but do not mention this detail in Table 4, which could lead to potential inconsistencies. To improve clarity and fairness, the authors should add explanations of any differences in the model architecture between datasets. This would help readers better understand the experimental setup and assess the fairness of the comparisons.


2. Although the authors claim to achieve "a balance between computational complexity and model expressivity" (line 820), the experiments related to computational efficiency (Tables 6,7) are not convincing enough. Adding runtime analysis on larger-scale datasets, such as ogbg-molhiv/ogbn-arxiv, would support these claims and better demonstrate the method's scalability.

**Questions:**

See Weaknesses

---

> ### Author Response · Authors · 2024-11-15
> **Rebuttal by Authors**
>
> We thank the reviewer for the constructive feedback!
>
> 1. We would like to clarify that both SIR-GCN models in Tables 4 and 5 largely employ the same architectural design with only a single GNN layer, as noted in Line 485 for Table 4 and Line 512 for Table 5. Appendix B2 presents a more detailed description of the architecture where the key difference between the two models lies in Table 5 including a graph readout function (since ogbg-molhiv is a graph property prediction task) whereas Table 4 does not (since ogbn-arxiv is a node property prediction task). The large discrepancy in parameter counts is primarily due to the input and output dimensions of each dataset. For ogbn-arxiv in Table 4, the input node features are 768-dimensional with a 40-dimensional output, while for ogbg-molhiv in Table 5, the input node features are only 174-dimensional with a scalar output. This difference in the input node feature dimension, stated in Lines 800 and 809, naturally affects the parameter count due to the Linear layer multiplying the input dimension by the hidden state dimension, even if they have the same architectural design. Specifically, a single Linear layer in ogbn-arxiv requires ~200,000 (768 x 256) parameters while a single Linear layer in ogbg-molhiv only requires ~53,000 (174 x 300) parameters. We also emphasize that, given the distinct nature and domains of these datasets (citation network vs molecular network), a comparison between parameter counts is not meaningful for fairness. Instead, we highlight that SIR-GCN can outperform larger models using fewer parameters in both datasets, particularly with LGGNN in Table 4 and GSN in Table 5. This highlights the utility and expressivity of SIR-GCN.
>
> 2. We have performed additional runtime analysis based on the reviewer's suggestion. The models below follow a similar architecture as reported in Appendix B2 for SIR-GCN using the same hyperparameters. The results still underscore how SIR-GCN effectively balances computational efficiency and model expressivity, particularly when compared against PNA which is also designed for uncountable node features. Notably, PNA resulted in an out-of-memory (OOM) error for ogbn-arxiv as it requires $O(|\mathcal{E}| \times d_\text{in})$ memory, where $|\mathcal{E}|$ is significantly larger for this graph and $d_\text{in} = 768$ as noted above in 1. On the other hand, PNA requires significantly longer (more than 50%) runtime compared to the other models in ogbg-molhiv, consistent with the results in Tables 6 and 7. Meanwhile, SIR-GCN requires approximately the same runtime as the other models across both datasets.
>
> |   Model   |     ogbn-arxiv     |    ogbg-molhiv    |
> | :-------: | :----------------: | :----------------: |
> |    GCN    | 0.0729s ± 0.0008s | 0.8029s ± 0.1171s |
> | GraphSAGE | 0.1161s ± 0.0039s | 0.8062s ± 0.1166s |
> |   GATv2   | 0.1006s ± 0.0022s | 0.6835s ± 0.1620s |
> |    GIN    | 0.0830s ± 0.0015s | 0.7574s ± 0.1620s |
> |    PNA    |        OOM        | 1.2786s ± 0.0664s |
> |  SIR-GCN  | 0.1265s ± 0.0039s | 0.7345s ± 0.1300s |

---

> ### Author Response · Authors · 2024-11-15
> **Rebuttal by Authors (cont.)**
>
> 3. We further highlight that the runtime analyses on synthetic datasets in Tables 6 and 7 provide a more controlled assessment of computational complexity, illustrating changes in model runtime as model complexity and problem size increase (*i.e.*, as $n$ and $c$ increase). This comparison allows for more meaningful insights into the scalability of each model in isolation. To further strengthen our claim, we have included an additional comparison of the asymptotic complexities of each model in Appendix C and in Line 825 to complement the results in Tables 6 and 7, specifically with regards to PNA which is also designed for uncountable node features but requires significantly longer runtime. The table further highlights how SIR-GCN has a computational complexity comparable to GCN, GraphSAGE, GAT, GATv2, and GIN while outperforming these models across all benchmarks. Notably, SIR-GCN also demonstrates a lower complexity than PNA, yet delivers superior performance across all datasets. These additional analyses underscore how SIR-GCN effectively balances computational efficiency and model expressivity, further demonstrating its novelty and practical utility.
>
> | Model       |                                                       Model Complexity                                                       |
> | ----------- | :--------------------------------------------------------------------------------------------------------------------------: |
> | GCN         |                           $O(V \times d_\text{out} \times d_\text{in} + E \times d_\text{out})$                           |
> | GraphSAGE   |                           $O(V \times d_\text{out} \times d_\text{in} + E \times d_\text{out})$                           |
> | GAT / GATv2 |                           $O(V \times d_\text{out} \times d_\text{in} + E \times d_\text{out})$                           |
> | GIN         |                                     $O(E \times d_\text{in} + V \times \texttt{MLP})$                                     |
> | PNA         |      $O(E \times d_\text{in}^2 + E \times d_\text{in} \times k + V \times d_\text{out} \times d_\text{in} \times k)$      |
> | SIR-GCN     | $O(V \times d_\text{hidden} \times d_\text{in} + E \times d_\text{hidden} + V \times d_\text{out} \times d_\text{hidden})$ |

---

> > ### Comment · Reviewer_YvUg · 2024-11-24
> >
> > Thank you for your clarification. It basically addresses my concern. I believe this paper offers some interesting perspectives on the representational capabilities of GNNs, and I will maintain my positive score.

---

### Official Review · Reviewer_rjgm · 2024-11-04

**Soundness:** 3
**Presentation:** 2
**Contribution:** 3
**Rating:** 6
**Confidence:** 4

**Summary:**

This paper investigates the representational capacity of Graph Neural Networks (GNNs) with uncountable node features and introduces an MLP-based MPNN named SIR-GCN, which generalizes to several popular GNN architectures. The authors demonstrate that for uncountable node features, it is possible to identify a soft-injective function corresponding to a specific pseudometric that quantifies dissimilarity in the node feature space. They then model this soft injective function using an MLP and design an architecture that maintains both anisotropic and isotropic properties. Experimental results highlight the model’s superiority across various scenarios, including cases with countable and uncountable node features, as well as graphs exhibiting heterophily.

**Strengths:**

1.	It is interesting to see how the author tackles the problem of uncountable node features using pseudo metric and soft-injective functions.
2.	The designed model has certain flexibility in terms of anisotropic and isotropic properties. The model architecture also generalizes easily to some popular GNNs.
3.	The experiment shows the SIR-GCN performance well on the Dictionary Lookup task and against other baseline models during the benchmarking test.

**Weaknesses:**

1.	Since the SIR-GCN can generalize to other GNNs, it would be better if the author can explain why it is hard for other GNNs to handle the problem of uncountable node features.
2.	It would be nice if the author can explain why there are some missing values in Table.
3.	For the graph heterophily experiment, it would be better if the author can use some real world datasets with different degrees of heterophily[1].

[1] Mao H, Chen Z, Jin W, Han H, Ma Y, Zhao T, Shah N, Tang J. Demystifying structural disparity in graph neural networks: Can one size fit all?. Advances in neural information processing systems, 2024.

**Questions:**

1.	According to the description of GAT, does GAT also preserve both anisotropic and isotropic?
2.	For Table 4, Can GAT or GraphSAGE achieve similar performance with more parameters?

---

> ### Author Response · Authors · 2024-11-15
> **Rebuttal by Authors**
>
> We thank the reviewer for the positive feedback!
>
> 1. The key novelty of SIR-GCN lies in its *anisotropic* and *dynamic* message function, allowing it to better handle uncountable node features. Unlike other GNNs which only employ *isotropic* or monotonic/linear transformations to neighborhood features, SIR-GCN is among the first MPNN instance to introduce *dynamic* transformations to neighborhood features that also account for the center node features. This design specifically enables SIR-GCN to learn the complex, nuanced relationships between pairs of neighboring nodes before aggregating them into a single feature. This is also key to how SIR-GCN handles uncountable node features, as explained in Line 527 of the conclusion. An illustration of this feature is further elaborated in 3 below.
>
> 2. Similar to the results presented in Corso et al. (2020), Table 3 also directly presents the available results from Dwivedi et al. (2023), Corso et al. (2020), and Tailor et al. (2021). The missing values indicate that these works have not considered the particular dataset for their model. This is added as a footnote in Table 3. We also emphasize that in addition to the datasets considered in Corso et al. (2020) for PNA, our paper further includes the WikiCS, PATTERN, and CLUSTER datasets from the Benchmarking GNNs (Dwivedi et al., 2023) as an additional evaluation.
>
> 3. We thank the reviewer for pointing us to the work of Mao et al. (2024). While the study does analyze real-world graphs with varying degrees of heterophily, the graphs considered are specific to node property prediction tasks. In contrast, GraphHeterophily is designed specifically for graph property prediction which makes the graphs considered in Mao et al. (2024) not directly compatible. Nonetheless, we emphasize that the directed graphs in GraphHeterophily are uniformly generated using DGL's `rand_graph` function, with class labels also uniformly assigned using PyTorch's `randint` function. These measures, highlighted in Line 775, ensure that the graphs are sufficiently diverse in terms of graph structure and heterophily degrees, supporting the robustness of the results.
>
> Furthermore, we would also like to highlight that while other GNNs fail in the simple task of GraphHeterophily, SIR-GCN excels in the task. Its remarkable performance is nevertheless expected, as explained in Line 430, since if node features are one-hot encodings of class labels, SIR-GCN with sum aggregation, $\boldsymbol{W_Q} = \boldsymbol{I}$, $\boldsymbol{W_K} = - \boldsymbol{I}$, $\sigma = \text{ReLU}$, and $\boldsymbol{W_R} = \boldsymbol{1}^\top$ can consistently produce accurate outputs, regardless of graph structure or heterophily degrees. This success may be attributed to the ReLU activation applied along edges, enabling SIR-GCN to "reason" along edges, based on the labels of pairs of neighboring nodes, and produce a *dynamic* message accounting for this learned relationship. This illustration further highlights the significance of *anisotropic* and *dynamic* message functions, underscoring the novelty of SIR-GCN as the first MPNN instance to satisfy this requirement.
>
> 4. The attention mechanism of the original GAT is *anisotropic* but not *dynamic* as highlighted by Brody et al. (2021). GATv2 addresses this limitation by making the attention mechanism *dynamic*. Nevertheless, both GAT and GATv2 still only linearly transform the messages (neighborhood features) as stated in Line 305. Thus, the features of the center node only affect the aggregated neighborhood features by determining the degree of contribution through the scalar attentional weight. In contrast, SIR-GCN is the first MPNN instance to leverage this idea and make the actual messages *anisotropic* and *dynamic*, rigorously grounded in theory. The experimental results, particularly in Table 3 where SIR-GCN significantly outperforms GAT across all datasets with the same parameter budget, underscore the significance of this idea and the novelty of SIR-GCN.
>
> References:
>
> Shaked Brody, Uri Alon, and Eran Yahav. How attentive are graph attention networks? arXiv preprint arXiv:2105.14491, 2021.
>
> Gabriele Corso, Luca Cavalleri, Dominique Beaini, Pietro Lio, and Petar Velickovic. Principal neighbourhood aggregation for graph nets. Advances in Neural Information Processing Systems, 33:13260-13271, 2020.
>
> Vijay Prakash Dwivedi, Chaitanya K Joshi, Anh Tuan Luu, Thomas Laurent, Yoshua Bengio, and Xavier Bresson. Benchmarking graph neural networks. Journal of Machine Learning Research, 24 (43):1-48, 2023.
>
> Haitao Mao, Zhikai Chen, Wei Jin, Haoyu Han, Yao Ma, Tong Zhao, Neil Shah, Jiliang Tang. Demystifying structural disparity in graph neural networks: Can one size fit all?. Advances in neural information processing systems, 2024.
>
> Shyam A Tailor, Felix L Opolka, Pietro Lio, and Nicholas D Lane. Do we need anisotropic graph neural networks? arXiv preprint arXiv:2104.01481, 2021.

---

> > ### Author Response · Authors · 2024-11-15
> > **Rebuttal by Authors (cont.)**
> >
> > 5. It may be possible to improve the performance of GAT and GraphSAGE in Table 4 with more parameters. However, as noted in Line 806, the results presented are taken directly from the OGB leaderboard. Extending GAT and GraphSAGE with more parameters for potential performance improvements would require significant tuning efforts and is beyond the scope of our work. Our primary focus is to highlight the performance of SIR-GCN relative to publicly released results in the leaderboard as stated in Lines 806 and 816.

---

> > > ### Comment · Reviewer_rjgm · 2024-11-25
> > >
> > > Thank you for the response. I tend to maintain my positive score.

---

### Official Review · Reviewer_NmsR · 2024-11-10

**Soundness:** 2
**Presentation:** 2
**Contribution:** 2
**Rating:** 3
**Confidence:** 2

**Summary:**

This paper extends PNA, which considers uncountable node features, by incorporating anisotropic and dynamic. The difference between PNA and the proposed SIR-GCN is the nonlinear mapping outside the message creation. To motivate this method, the authors introduce the concept of soft-injective function. This paper shows some existing methods are variants of the proposed SIR-GCN. Evaluations on synthetic and real datasets demonstrate its effectiveness.

**Strengths:**

- The uncountable feature is an important topic in GNN.
- The writing and organization are good to follow.
- The insight of existing methods under the framework of SIR-GCN is interesting.

**Weaknesses:**

- The novelty seems weak. Both the anisotropic and dynamic are not novel.  This paper can be seen as a combination of PNA and GATv2.
- The motivation and the proposed SIR-GCN are not closely connected. It is not clear the connection between the soft-injective function, dynamic transformation, and anisotropic message.
- The description of the GraphHeterophily is not clear. Thus,  it is not obvious why the proposed SIR-GCN significantly outperforms existing ones.
- The derivation from Eq. 15 to Eq. 16 seems incorrect. First, the definition of $A$ is not given. Secondly, the anisotropic of GAT is on the edge weight, while that of Eq. 16 is on the message. It is not obvious.
- Figure 2 is not described clearly. What is the meaning of the horizontal and vertical coordinates? Why the contour of MLP is as in Figure 2(c) and 2(d).
- The evaluations are not convincing. Firstly, the ablation study and illustrative examples are not given. So, the effect of the proposed SIR-GCN is not justified. Secondly, it is not knowns whether the proposed SIR-GCN can be applied to complex models, whose performance is higher.

**Questions:**

See weakness.

---

> ### Author Response · Authors · 2024-11-15
> **Rebuttal by Authors**
>
> We thank the reviewer for the constructive insights!
>
> 1. We would first like to clarify that our work is not an extension of Corso et al. (2020) but rather a novel approach to handling uncountable node features within the MPNN framework. Unlike PNA whose key features are multiple aggregators and scalers, SIR-GCN only employs a single aggregator. Consequently, it is also computationally efficient, requiring only an activation function (linear complexity) along edges, unlike PNA, which requires a full linear layer (quadratic complexity) as shown in Table 8 of Appendix C. Despite its lower computational requirement, SIR-GCN still achieves superior performance, consistently outperforming PNA across all benchmarks spanning both countable and uncountable node features, highlighting its efficiency, expressivity, and practical utility. We further note that unlike PNA whose *anisotropic* nature is simply a direct consequence of applying its main theoretical result of multiple aggregators and scalers within the MPNN framework (Section 2.3 of Corso et al., 2020), our work presents a rigorous theoretical foundation for the *anisotropic* nature of SIR-GCN in Line 216, highlighting its novel contribution to GNN research.
>
> Moreover, while Brody et al. (2021) explored *anisotropic* and *dynamic* functions in the context of GAT attention mechanisms, our work is the first to apply these principles specifically to message functions within the broader MPNN framework. This work contributes to GNN literature by providing a rigorous theoretical foundation (Lines 192 and 216) and extensive empirical results on how this key innovation enables SIR-GCN to capture complex, nuanced relationships between pairs of neighboring nodes prior to aggregation, allowing it to better handle uncountable node features. The significance of this contribution is evident in the experimental results, particularly in Table 3 where SIR-GCN consistently outperforms all other GNNs (including GAT) with the same parameter budget. This is also evident in Table 2 where SIR-GCN successfully accomplished the task when all other GNNs failed, as elaborated in 3 below.
>
> Overall, our work offers a novel perspective into the representational capability of GNNs, distinct from Corso et al. (2020) and Brody et al. (2021), especially in the problem of uncountable node features, by demonstrating that using only a single aggregator can already substantially improve the representational capability of GNNs. The SIR-GCN, through its use of *anisotropic* and *dynamic* message functions within the MPNN framework, introduces a fundamentally novel approach to this problem, distinguishing it from existing models like PNA and GATv2 whose specific utility and novelty differs from our work. Notably, the unique design of SIR-GCN directly addresses several limitations of existing GNNs, as demonstrated by illustrative examples on synthetic datasets and model performance on benchmark datasets. These findings establish SIR-GCN as a significant and novel contribution to advancing GNN research.
>
> 2. The connection between the motivation and the proposed SIR-GCN is firmly established through the theoretical foundation provided by *pseudometrics* and *soft-injective* functions. Corollary 1 guarantees the existence of a *soft-injective* hash function $G$ and *soft-injective* feature map $g$ given a *pseudometric* $d$ on $\mathcal{H}$ and *pseudometric* $D$ on bounded equinumerous *multisets* of $\mathcal{H}$. From this result, two necessary properties of the *soft-injective* message function $g$ emerge, as elaborated in Line 189 onward, which directly inform the design of SIR-GCN. As clarified in Line 192, for arbitrary *pseudometrics* $d$, the corresponding *soft-injective* message function $g$ must be *dynamic* or non-linear. This insight motivates the use of an MLP to model the *dynamic* nature of $g$, similar to Brody et al. (2021). From Line 216, we further show that the *soft-injective* message function $g$ must also adapt to each node independently. This insight then motivates integrating the features of the center node into $g$, consequently making it *anisotropic*, to avoid the impracticality of designing distinct message functions for each node in large graphs. Combining these two properties, SIR-GCN is the first MPNN instance to utilize a *soft-injective* message function that is both *anisotropic* and *dynamic*. This connection between the theoretical motivation and the architectural design ensures that SIR-GCN is not just empirically performant but also rigorously grounded in theory, demonstrating the significance and novelty of our approach.
>
> References:
>
> Shaked Brody, Uri Alon, and Eran Yahav. How attentive are graph attention networks? arXiv preprint arXiv:2105.14491, 2021.
>
> Gabriele Corso, Luca Cavalleri, Dominique Beaini, Pietro Lio, and Petar Velickovic. Principal neighbourhood aggregation for graph nets. Advances in Neural Information Processing Systems, 33:13260-13271, 2020.

---

> ### Author Response · Authors · 2024-11-15
> **Rebuttal by Authors (cont.)**
>
> 3. The GraphHeterophily dataset is designed to test the ability of models to reason about heterophilous relationships in directed graphs. The graphs are uniformly generated using DGL's `rand_graph` function with each node uniformly assigned one of $c$ class labels using PyTorch's `randint` function. This approach ensures diversity in graph structures and degrees of heterophily, making the dataset robust for evaluation. Detailed descriptions are provided in Appendix B1. The task is then to count the total number of directed edges in each graph connecting nodes with different class labels, as clarified in Line 417. A sample graph is illustrated in Fig. 5, where four nodes (labeled A or B) are connected by six directed edges. Models must then correctly identify/count the four edges (highlighted in blue) that connect nodes with distinct labels.
>
> The results in Table 2 show that SIR-GCN consistently achieves near-zero MSE loss for this simple illustrative task while other models (including PNA and GATv2) obtained large losses. This performance is nevertheless expected, as explained in Line 431, since if node features are one-hot encodings of class labels, SIR-GCN with sum aggregation, $\boldsymbol{W_Q} = \boldsymbol{I}$, $\boldsymbol{W_K} = - \boldsymbol{I}$, $\sigma = \text{ReLU}$, and $\boldsymbol{W_R} = \boldsymbol{1}^\top$ can consistently produce accurate outputs, regardless of graph structure or heterophily degrees. The key to this success lies in the *anisotropic* and *dynamic* message function enabled by the ReLU activation along edges, which allows SIR-GCN to "reason" based on pairs of neighboring node labels and generate nuanced, context-aware messages. It is worth noting that while GATv2 can also somewhat "reason" along edges due to its *dynamic* attention mechanism, its reliance on attentional (softmax) aggregation, which fails to make sharp decisions (Velickovic, 2024) and preserve graph structure as noted in Line 433, hinders its performance. This distinction further highlights the flexibility of SIR-GCN in handling such challenges. Overall, this task underscores the importance of *anisotropic* and *dynamic* message functions (in contrast to attention mechanism), demonstrating SIR-GCN as the first MPNN instance to meet these requirements, further solidifying its contribution and novelty with respect to existing GNNs.
>
> 4. We thank the reviewer for pointing out the lack of definition for $\boldsymbol{A}$. In response, we have changed $\boldsymbol{W_R} = \boldsymbol{a_\text{GAT}^\top}$ for clarity. To clarify, Eq. 16 simply demonstrates how the unnormalized attention mechanism in Eq. 15 for GATv2 may be interpreted as the contextualized message in the SIR-GCN model, as mentioned in Line 313. We do not intend to suggest that Eq. 16 is equivalent to GAT. Instead, the illustration shows how the concept of *anisotropic* and *dynamic* functions in the attention mechanism of GATv2 are adapted to message functions in SIR-GCN. This aligns with the explanation above in 1. Furthermore, we also emphasize that only the attention weights in GATv2 are *anisotropic* and *dynamic*, its message functions are still only linearly transformed, which can limit its expressivity, as discussed in Line 305. In contrast, SIR-GCN introduces a contextualized, non-linear transformation to the message functions, improving its ability to capture complex relationships. The significance of this contribution is evident in the experimental results, particularly in Table 3 where SIR-GCN significantly outperforms GAT across all datasets with the same parameter budget. Nevertheless, in Line 314, we clarify how GATv2, up to a normalizing constant, can be obtained from SIR-GCN by selecting the appropriate parameters: $\boldsymbol{W_Q} = \begin{bmatrix} \boldsymbol{W_{Q,\text{GAT}}} \\\\ \boldsymbol{0} \end{bmatrix}$, $\boldsymbol{W_K} = \begin{bmatrix} \boldsymbol{W_{K,\text{GAT}}} \\\\ \boldsymbol{W_{K,\text{GAT}}} \end{bmatrix}$, $\sigma\left(\begin{bmatrix} \boldsymbol{h_1} \\\\ \boldsymbol{h_2} \end{bmatrix}\right) = \exp\left(\boldsymbol{a_\text{GAT}^\top} ~ \text{LeakyReLU}\left(\boldsymbol{h_1}\right)\right) \cdot \boldsymbol{h_2}$, and $\boldsymbol{W_R} = \boldsymbol{I}$.
>
> References:
>
> Petar Velickovic, Christos Perivolaropoulos, Federico Barbero, and Razvan Pascanu. softmax is not enough (for sharp out-of-distribution). arXiv preprint arXiv:2410.01104, 2024.

---

> ### Author Response · Authors · 2024-11-15
> **Rebuttal by Authors (cont.)**
>
> 5. The horizontal and vertical axes of Fig. 2 represent the scalar neighborhood node features $\boldsymbol{h_{v_1}}$ and $\boldsymbol{h_{v_2}}$ of node $u$, where $v_1$ and $v_2$ are its neighbors. Line 193 provides context where these features represent zero-mean scores for anomaly detection. Fig. 2 illustrates how these features are transformed and aggregated using different *soft-injective* message functions. As clarified in Line 192, for arbitrary *pseudometrics* $d$, the corresponding *soft-injective* message function $g$ must be *dynamic* or non-linear. This insight motivates the use of an MLP to model the *dynamic* nature of the *soft-injective* message function, similar to Brody et al. (2021). Figs. 2(c) and 2(d) then highlight how MLPs may be used to model *anisotropic* and *dynamic* (*soft-injective*) message functions within the SIR-GCN model, aligning with the theoretical motivation discussed in the paper.
>
> 6. The results from the DictionaryLookup and GraphHeterophily synthetic datasets serve as key illustrative examples of the practical utility and novelty of SIR-GCN. In the DictionaryLookup task, both GATv2 and SIR-GCN achieved perfect accuracy, highlighting the utility of a *dynamic* attentional or relational mechanism in capturing the relationships between *query* and *key* nodes, as explained in Line 413. In contrast, the GraphHeterophily dataset clearly exposes the limitations of existing GNNs (including GATv2), as only SIR-GCN achieved near-zero MSE loss while all other GNNs obtained large errors, as detailed above in 3. This dataset thus underscores the utility and novelty of SIR-GCN with its *anisotropic* and *dynamic* message functions (in contrast to attention mechanism).
>
> Furthermore, the results on benchmark datasets further highlight the superior performance of SIR-GCN over existing GNNs in more complex, real-world problems across various domains. In these evaluations, the SIR-GCN models implemented follow standard model design for GNNs, as outlined in Appendix B2. Specifically, we directly replace the GNN component (from GCN, GraphSAGE, GAT, GIN, and PNA) with SIR-GCN. Since we focused on this single replacement, performing an ablation study is neither appropriate nor feasible, similar to how no ablation study was conducted by Brody et al. (2021) when GAT was replaced with GATv2. Nevertheless, Section 4 provides a detailed mathematical discussion of how modifying specific parameters in SIR-GCN may recover conventional GNNs, whose performance is already included in our results. Hence, the reported results and performance improvements are solely attributed to the novel aspects of SIR-GCN. We are open to suggestions for further improving the results section.
>
> Finally, we emphasize that SIR-GCN can be easily integrated into more complex frameworks, such as grouped reversible residual connections (Li et al., 2021) and graph stochastic attention (Miao et al., 2022), to further improve performance. Previous works, particularly in Li et al. (2021) and Miao et al. (2022), have demonstrated that any GNN backbone, such as GCN, GraphSAGE, GAT, GIN, and PNA, can be seamlessly and easily incorporated into their frameworks to employ additional/advanced techniques for enhanced performance. While this also holds for SIR-GCN, exploring such integrations with SIR-GCN is beyond the scope of this paper and left for future works as explained in the conclusion, as it primarily focuses on introducing the key contributions and foundations of SIR-GCN.
>
> References:
>
> Shaked Brody, Uri Alon, and Eran Yahav. How attentive are graph attention networks? arXiv preprint arXiv:2105.14491, 2021.
>
> Guohao Li, Matthias Muller, Bernard Ghanem, and Vladlen Koltun. Training graph neural networks with 1000 layers. In International Conference on Machine Learning, pp. 6437-6449. PMLR, 2021.
>
> Siqi Miao, Mia Liu, and Pan Li. Interpretable and generalizable graph learning via stochastic attention mechanism. In International Conference on Machine Learning, pp. 15524-15543. PMLR, 2022.

---

> > ### Author Response · Authors · 2024-11-22
> > **Additional Experiments**
> >
> > 7. Based on the reviewer's feedback, we have conducted additional experiments in Appendix D to further highlight the utility and novelty of SIR-GCN as the first MPNN instance to theoretically and empirically justify the use of *anisotropic* and *dynamic* message functions. Specifically, we consider SIR-GCN (*static*), which uses linear messages by setting $\sigma$ as identity and $\boldsymbol{W_R} = \boldsymbol{I}$, and SIR-GCN (*isotropic*), which removes the dependency of messages on center node features by setting $\boldsymbol{W_Q} = \boldsymbol{0}$. Although SIR-GCN achieves lower accuracy on WikiCS compared to the two simpler SIR-GCNs (*static* and *isotropic*), this result is consistent with the dataset's characteristics. As noted by Dwivedi et al. (2023), WikiCS is a single-graph dataset with denser node neighborhoods and shorter average path lengths, which can make more expressive models like SIR-GCN prone to overfitting and oversmoothing. Thus, the simpler SIR-GCNs are naturally less expressive and achieve higher accuracies for this small dataset. In contrast, on larger and more complex datasets such as PATTERN, CLUSTER, MNIST, CIFAR10, and ZINC, SIR-GCN consistently outperforms both the simpler SIR-GCNs and conventional GNNs. This underscores the strong utility of **both** *anisotropic* and *dynamic* message functions in improving GNN representational capability. Overall, these additional results highlight the novelty of SIR-GCN and further confirm the theoretical and practical contributions of our work in advancing GNN research.
> >
> > | Model                   |  WikiCS (↑)  | PATTERN (↑) |  CLUSTER (↑)  |   MNIST (↑)   |  CIFAR10 (↑)  |    ZINC (↓)    |
> > | :---------------------- | :------------: | :-----------: | :------------: | :------------: | :------------: | :-------------: |
> > | SIR-GCN (*static*)    | 78.52 ± 0.57 | 85.72 ± 0.02 | 61.90 ± 0.25 | 95.65 ± 0.84 | 50.09 ± 3.20 | 0.334 ± 0.014 |
> > | SIR-GCN (*isotropic*) | 78.73 ± 0.63 | 85.74 ± 0.03 | 62.60 ± 0.38 | 97.44 ± 0.11 | 68.88 ± 0.27 | 0.281 ± 0.024 |
> > | SIR-GCN                 | 78.06 ± 0.66 | 85.75 ± 0.03 | 63.35 ± 0.19 | 97.90 ± 0.08 | 71.98 ± 0.40 | 0.278 ± 0.024 |
> >
> > References:
> >
> > Vijay Prakash Dwivedi, Chaitanya K Joshi, Anh Tuan Luu, Thomas Laurent, Yoshua Bengio, and Xavier Bresson. Benchmarking graph neural networks. Journal of Machine Learning Research, 24 (43):1-48, 2023.

---

> > > ### Comment · Reviewer_NmsR · 2024-11-22
> > >
> > > Thanks for the feedback from the authors. Although their comments alleviate some of my concerns, they do not clarify my concerns about the novelty. I also believe it is very similar to GATv2. GATv2 explores anisotropic and dynamic functions on edge weight function, while this paper explores them on the message function.  Besides, the performance improvement on real-world datasets is incremental and the extension to complex methods is not provided. Thus, I tend to keep my ratings.

---

> ### Author Response · Authors · 2024-11-22
> **Additional Clarifications**
>
> We thank the reviewer for the follow-up feedback.
>
> 1. We would like to clarify that the novelty of our work is **two-fold**. First, we are the first to provide a **novel theoretical framework** based on *pseudometrics* and *soft-injective* functions to enhance the representational capability of GNNs. This theoretical contribution addresses a critical area in GNN research on uncountable node features, providing an **alternative perspective from the multiple aggregators of PNA**. Second, we further provide a detailed discussion bridging this novel theoretical framework into the MPNN framework, resulting in the SIR-GCN, which is **empirically demonstrated to outperform conventional GNNs across several synthetic and benchmark datasets**.
>
> 2. While GATv2 explores the *anisotropic* and *dynamic* properties of GAT **attention weights**, its original motivation for using attention mechanisms are nevertheless **heuristic**. The authors **do not provide a discussion on the representational capability of GATv2**. Moreover, we also highlight the **limitations of both GAT and GATv2**, attributed to their **linear transformation** of neighborhood features and use of **softmax** (Velickovic et al., 2024), which limits their expressivity. In contrast, our work fundamentally differs in context. In particular, our focus is on applying *anisotropic* and *dynamic* properties directly to the **message function**. This allows SIR-GCN to **effectively handle uncountable node features**, as highlighted in Line 531. Critically, we are the first to provide a **rigorous theoretical justification for this approach**, specifically to **enhance GNN representational capability**. This is complemented by **extensive empirical results** showing the contribution of this novel idea.
>
> 3. We respectfully disagree with the assessment that performance improvements on real-world datasets are incremental. **SIR-GCN consistently outperforms conventional GNNs** (including GCN, GraphSAGE, GAT, and GIN) by a **significant margin**, as shown in Tables 3, 4, and 5. To further put this into perspective, **SIR-GCN also achieves substantial improvements over PNA** in real-world datasets like MNIST, CIFAR10, ZINC, and ogbg-molhiv while employing a **simpler and computationally efficient design**, as highlighted in Appendix C, **further highlighting the utility and novelty of SIR-GCN**.
>
> 4. We would like to emphasize that the complex frameworks, such as grouped reversible residual connections (Li et al., 2021) and graph stochastic attention (Miao et al., 2022), are introduced simply to provide **concrete steps for future works** to build upon our theoretical and empirical results. This analysis is beyond the scope of the current work, similar to **foundational works** such as GATv2 and PNA, since our primary objective is to **demonstrate the core contributions of our proposed SIR-GCN** without employing additional tricks or techniques, as explicitly stated in Line 381. This ensures a fair evaluation where performance gains are **solely attributed to the key features of SIR-GCN**, in line with the reviewer's earlier remark of **justifying the effect of SIR-GCN**.
>
> 5. Overall, we firmly believe that our work makes a **significant and novel contribution to GNN research** by providing a **comprehensive theoretical perspective for handling uncountable node features**, supported by empirical validation, and a **practical and efficient SIR-GCN that consistently outperforms conventional GNNs**. Together, these contributions provide a ***new, relevant, and impactful* advancement toward understanding GNN representational capabilities**.
>
> References:
>
> Guohao Li, Matthias Muller, Bernard Ghanem, and Vladlen Koltun. Training graph neural networks with 1000 layers. In International Conference on Machine Learning, pp. 6437-6449. PMLR, 2021.
>
> Siqi Miao, Mia Liu, and Pan Li. Interpretable and generalizable graph learning via stochastic attention mechanism. In International Conference on Machine Learning, pp. 15524-15543. PMLR, 2022.
>
> Petar Velickovic, Christos Perivolaropoulos, Federico Barbero, and Razvan Pascanu. softmax is not enough (for sharp out-of-distribution). arXiv preprint arXiv:2410.01104, 2024.

---

> > ### Comment · Reviewer_NmsR · 2024-11-24
> >
> > Thanks for your further responses.
> > 1. I acknowledge the novel theoretical framework which results in the proposed SIR-GCN.
> > 2. GATv2 also provides a discussion on the representational capability as in Theorems 1 and 2 in that paper.
> > 3. The performance improvement should be based on the SOTA instead of the basic GNNs. Thus, it is incremental.
> > 4. Being universal to other complex models may demonstrate the impact of the proposed theory on the graph machine learning field. Thus, it is important to me.
> >
> > According to the above considerations, I tend to keep my ratings.

---

> ### Author Response · Authors · 2024-11-25
> **Alignment with Prominent Foundational Works in GNN**
>
> We thank the reviewer for the additional feedback. We would like to clarify how our work aligns with prominent foundational works in GNN.
>
> 1. The theoretical discussions by Brody et al. (2021) only focus on the **representational capability of the GAT attention mechanism**. Specifically, they only analyzed the ***difference* in representational capability between *static* (Theorem 1) and *dynamic* (Theorem 2) GAT attention**, as explicitly written in their theorem statements. In contrast, our theoretical discussions focus on the **broader representational capability of GNNs with a specific focus on uncountable node features** which is **not addressed by Brody et al. (2021)**.
>
> 2. In **Xu et al. (2018), Corso et al. (2020), Brody et al. (2021)**, and other prominent foundational works in GNN, they only compared the performance of their proposed *foundational models* (GIN, PNA, GATv2) to **other *foundational GNNs*** (GCN, GraphSAGE, GAT). Notably, these ***foundational models* are not designed to explicitly achieve state-of-the-art (SOTA) performance**. Hence, these works do not compare the performance of their model to SOTA models to ensure a **fair evaluation** where performance improvements are solely attributed to the **key features of the message-passing architecture**. Following these *foundational works*, our results also compare the performance of our proposed *foundational model* SIR-GCN only to *foundational GNNs* (GCN, GraphSAGE, GAT, GATv2, GIN, PNA) to highlight the **performance improvement attributed to the key message-passing features of SIR-GCN**. These results also underscore the practical utility of ***replacing conventional foundational GNNs* with SIR-GCN in existing models**. Furthermore, given the ***foundational* nature of SIR-GCN which is (similar to other *foundational models*) not designed to explicitly achieve SOTA**, a direct comparison of performance relative to SOTA would **not be meaningful nor comparable** and would **require *significant* model architecture tuning** since these models employ **several additional techniques that significantly increase their performance**.
>
> 3. **Xu et al. (2018), Corso et al. (2020), Brody et al. (2021)**, and other prominent foundational works in GNN only provide a **theoretical analysis of their key contributions based on the MPNN framework**. It is ***subsequent and separate* works** that extended their results to other complex frameworks as this **required additional analysis that is *beyond the scope of the original paper***. Similar to these *foundational works*, our *foundational paper* also presents a **comprehensive theoretical framework based on MPNN**. Extending our results to other frameworks also requires ***significant* additional analysis** that is ***beyond the scope of our paper*** and warrants a **comprehensive discussion in a *separate* and dedicated work**. Nevertheless, our novel theoretical framework based on MPNN already provides a ***new, relevant, and impactful* advancement toward understanding GNN representational capabilities**.
>
> 4. Overall, our work, *in its current form*, already provides a **significant and novel contribution to GNN literature** on the problem of uncountable node features. Furthermore, as a ***foundational paper*** introducing the *foundational GNN model* SIR-GCN, it also already **includes the standard and key discussions and comparisons** in prominent foundational works in GNN.
>
> References:
>
> Shaked Brody, Uri Alon, and Eran Yahav. How attentive are graph attention networks? arXiv preprint arXiv:2105.14491, 2021.
>
> Gabriele Corso, Luca Cavalleri, Dominique Beaini, Pietro Lio, and Petar Velickovic. Principal neighbourhood aggregation for graph nets. Advances in Neural Information Processing Systems, 33:13260–13271, 2020.
>
> Keyulu Xu, Weihua Hu, Jure Leskovec, and Stefanie Jegelka. How powerful are graph neural networks? arXiv preprint arXiv:1810.00826, 2018.

---

### Author Response · Authors · 2024-11-17
**Clarifications on Novelty and Significance**

We would like to further clarify the novelty and significance of our work, especially in relation to Velickovic et al. (2017), Brody et al. (2021), and Corso et al. (2020).

First, while Velickovic et al. (2017) introduced the original GAT by integrating attention mechanisms into GNNs, their work primarily focused on heuristic and empirical validation. It lacked a rigorous theoretical analysis of how attention mechanisms impact the representational capability of GNNs. Brody et al. (2021) later extended GAT by specifically analyzing the role of *dynamic* attention mechanisms, providing theoretical insights into their utility. However, their contributions were still limited to refining the intuition behind GAT and did not explore its implications for handling uncountable node features or the broader representational capability of GNNs. Moreover, given the focus on attention mechanisms, both studies simply employ linear transformations to the actual messages.

Meanwhile, Corso et al. (2020) specifically tackled the problem of uncountable node features by providing theoretical justifications for using multiple aggregators and scalers in GNNs to enforce injectivity for graph isomorphism tasks, resulting in the PNA. Nonetheless, the *anisotropic* nature of PNA emerges as a consequence of applying the theoretical result heuristically within a linear/*static* MPNN framework as we noted in Line 343, rather than through theoretical insight. In fact, our analysis in Line 332 reveals a critical limitation in the *anisotropic* nature of PNA, specifically in the limited influence of center node features on the aggregated neighborhood features.

In contrast, our work fundamentally diverges from these prior studies by introducing a **novel and comprehensive perspective** on the representational capability of GNNs with uncountable node features. Specifically, we are the first to introduce **new theoretical results**, based on *pseudometrics* and *soft-injective* functions, demonstrating how GNNs can preserve their representational capability even by relaxing the strict injective and metric constraints in previous works, such as Corso et al. (2020). Our main theoretical result in Corollary 1 directly leads to the emergence of the *anisotropic* and *dynamic* properties of *soft-injective* message functions, as outlined in Lines 192 and 216. While these properties have been studied independently by Brody et al. (2021), their presence here only serves to underscore how existing works complement and validate our theoretical findings. Crucially, our work is the first to theoretically justify how these properties, when applied to message functions (in contrast to attention mechanisms), enable GNNs, specifically SIR-GCN, to effectively handle uncountable node features, as highlighted in Line 531. To support our theoretical findings, we also provide **intuitive illustrations** through two synthetic datasets, demonstrating the limitations of existing GNNs in simple node and graph property prediction tasks and how SIR-GCN, with its novel design, specifically addresses these weaknesses, offering an intuitive understanding of the practical utility of our theoretical results. Additionally, our framework is validated by **extensive experimental results** on benchmark datasets, where the theoretical concepts and intuition presented concretely translate to SIR-GCN consistently outperforming conventional GNNs across diverse tasks and domains, highlighting the significance of our theoretical results. Overall, our work introduces a novel theoretical foundation for a key problem in GNNs that is supported by both intuition and experimental results, thereby advancing our understanding of the representational capabilities of GNNs.

References:

Shaked Brody, Uri Alon, and Eran Yahav. How attentive are graph attention networks? arXiv preprint arXiv:2105.14491, 2021.

Gabriele Corso, Luca Cavalleri, Dominique Beaini, Pietro Lio, and Petar Velickovic. Principal neighbourhood aggregation for graph nets. Advances in Neural Information Processing Systems, 33:13260–13271, 2020.

Petar Velickovic, Guillem Cucurull, Arantxa Casanova, Adriana Romero, Pietro Lio, and Yoshua Bengio. Graph attention networks. arXiv preprint arXiv:1710.10903, 2017.

---

### Author Response · Authors · 2024-12-02
**Summary of Rebuttal Discussions**

We sincerely thank the reviewers for their time and valuable feedback!

In summary, our paper introduces a novel theoretical framework based on *pseudometrics* and *soft-injective* functions for understanding the representational capabilities of GNNs, with a specific focus on uncountable node features. Through rigorous theoretical analysis, we highlight that message functions in the MPNN framework must be *anisotropic* and *dynamic*. We then translate this theoretical insight into our proposed SIR-GCN, which is the first MPNN instance to satisfy these properties. Empirically, we also demonstrate how this key feature allows SIR-GCN to generalize and consistently outperform existing foundational GNNs, including GCN, GraphSAGE, GAT, GIN, and PNA, across various synthetic and benchmark datasets spanning diverse domains. Our work thus provides new, relevant, and impactful theoretical and empirical advancements toward understanding GNN representational capabilities.

Notably, our work fundamentally differs from Brody et al. (2021) by offering a comprehensive theoretical analysis of the broader representational capabilities of GNNs, extending beyond the GAT attention mechanism investigated by Brody et al. (2021). While GATv2 applies *anisotropic* and *dynamic* functions on its attention mechanism, its messages nevertheless remain linear, potentially limiting its expressivity. In contrast, motivated by our novel theoretical results, the *anisotropic* and *dynamic* messages of SIR-GCN are explicitly designed to boost GNN representational capability. Similarly, while Corso et al. (2020) also examined the representational capabilities of GNNs with uncountable node features, our theoretical framework is fundamentally distinct in being the first to *softly* relax the injective and metric requirements of prior works. This approach allows SIR-GCN to achieve computational efficiency with a single aggregator while still outperforming PNA and GATv2 despite its simple design. These key differences in scope and approach underscore the novelty and significance of our work.

In response to Reviewer NmsR, we would also like to clarify that, consistent with prominent foundational works in GNN such as Xu et al. (2018), Corso et al. (2020), and Brody et al. (2021), our foundational paper also focuses on introducing novel theoretical insights and developing a foundational GNN model within the MPNN framework. Extending these theoretical results to more complex frameworks, while important, involves significant additional theoretical analysis and experimentation that warrants a comprehensive discussion in a separate and dedicated study, similar to how separate and subsequent papers extended the results of Xu et al. (2018) to higher-order WL tests. Furthermore, these foundational GNN works also primarily compared their proposed models (*e.g.*, GIN, GATv2, PNA) to other foundational GNNs (*e.g.*, GCN, GraphSAGE, GAT) to isolate and highlight performance improvements solely attributed to the key features of their model. Following these works, we only directly compare SIR-GCN to existing foundational GNNs but do not extend the comparison to state-of-the-art models, as these incorporate additional techniques that significantly enhance their performance, making such a comparison neither fair nor meaningful. This experimental analysis would also require substantial model tuning and experimentation that also deserves a separate study to ensure rigor. Nonetheless, we emphasize that our work, in its current form, already makes a significant contribution to the GNN literature by addressing the important problem of uncountable node features and providing novel theoretical and empirical insights. Furthermore, as a foundational work, it already includes the key discussions and comparisons standard in prominent foundational GNN works.

References:

Shaked Brody, Uri Alon, and Eran Yahav. How attentive are graph attention networks? arXiv preprint arXiv:2105.14491, 2021.

Gabriele Corso, Luca Cavalleri, Dominique Beaini, Pietro Lio, and Petar Velickovic. Principal neighbourhood aggregation for graph nets. Advances in Neural Information Processing Systems, 33:13260-13271, 2020.

Keyulu Xu, Weihua Hu, Jure Leskovec, and Stefanie Jegelka. How powerful are graph neural networks? arXiv preprint arXiv:1810.00826, 2018.

---

### Note · Authors · 2025-02-04

I have read and agree with the venue's withdrawal policy on behalf of myself and my co-authors.